# Discovery of a small molecule that selectively destabilizes Cryptochrome 1 and enhances life span in *p53* knockout mice

Seref Gul [1,12], Yasemin Kubra Akyel [2,13], Zeynep Melis Gul[3], Safak Isin [3], Onur Ozcan [3], Tuba Korkmaz[4], Saba Selvi [4], Ibrahim Danis[5,6], Ozgecan Savlug Ipek[7,8], Fatih Aygenli[4], Ali Cihan Taskin[9], Büşra Aytül Akarlar[3], Nurhan Ozlu [3], Nuri Ozturk [4], Narin Ozturk[2], Durişehvar Özer Ünal[5,6], Mustafa Guzel[7,10], Metin Turkay [11], Alper Okyar[2] & Ibrahim Halil Kavakli [1,3] ✉

Cryptochromes are negative transcriptional regulators of the circadian clock in mammals. It is not clear how reducing the level of endogenous CRY1 in mammals will affect circadian rhythm and the relation of such a decrease with apoptosis. Here, we discovered a molecule (M47) that destabilizes Crypto-chrome 1 (CRY1) both in vitro and in vivo. The M47 selectively enhanced the degradation rate of CRY1 by increasing its ubiquitination and resulted in increasing the circadian period length of U2OS *Bmal1*-d*Luc* cells. In addition, subcellular fractionation studies from mice liver indicated that M47 increased degradation of the CRY1 in the nucleus. Furthermore, M47-mediated CRY1 reduction enhanced oxaliplatin-induced apoptosis in Ras-transformed *p53* null fibroblast cells. Systemic repetitive administration of M47 increased the median lifespan of *p53*−/− mice by ~25%. Collectively our data suggest that M47 is a promising molecule to treat forms of cancer depending on the *p53* mutation.

The circadian clock is responsible for generating a 24-h rhythm that allows physiology and behavior to adjust daily environmental changes[1]. The circadian clock governs various biological functions, including hormone release and sleep-wake cycles[2]. Consequently, a natural malfunction of the circadian clock is associated with a variety of diseases, including mental disorders, altered metabolism, sleep disorders, obesity, cardiovascular diseases, and diabetes[3–5].

Several proteins that play a role in both positive and negative transcriptional-translational feedback loops (TTFL) are required for maintaining and generating the circadian clock in mammals. Among them, circadian locomotor output cycles caput (CLOCK) and aryl hydrocarbon receptor nuclear translocator-like (BMAL1) form het-erodimer and bind E-box (CACGTG) in the promotor of the clock-controlled genes (CCGs). Then, they initiate transcription of CCGs

[1]Department of Chemical and Biological Engineering, Koc University, 34450 Sariyer-Istanbul, Turkey. [2]Faculty of Pharmacy, Department of Pharmacology, İstanbul University, TR-34116 Beyazit-Istanbul, Turkey. [3]Department of Molecular Biology and Genetics, Koc University, İstanbul, Turkey. [4]Department of Molecular Biology and Genetics, Gebze Technical University, Gebze, 41400 Kocaeli, Turkey. [5]Faculty of Pharmacy, Department of Analytical Chemistry, İstanbul University, TR-34116 Beyazit-Istanbul, Turkey. [6]İstanbul University Drug Research and Application Center (ILAM), TR-34116 Beyazıt-Istanbul, Turkey. [7]Regenerative and Restorative Medicine Research Center (REMER), İstanbul Medipol University, Kavacik Campus, Kavacik-Beykoz/Istanbul 34810, Turkey. [8]Department of Chemistry, Graduate School of Natural and Applied Sciences, Yildiz Technical University, Besiktas/Istanbul 34349, Turkey. [9]Animal Research Facility, Research Center for Translational Medicine, Koc University, Rumelifeneri yolu, 34450 Sariyer-Istanbul, Turkey. [10]International School of Medicine, Department of Medical Pharmacology, Kavacik Campus, İstanbul Medipol University, Kavacik-Beykoz/Istanbul 34810, Turkey. [11]Department of Industrial Engineering, Koc University, Istanbul, Turkey. [12]Present address: Department of Biology, Biotechnology Division, İstanbul University, TR-34116 Beyazit-Istanbul, Turkey. [13]Present address: School of Medicine, Department of Medical Pharmacology, Istanbul Medipol University, Istanbul, Turkey. ✉e-mail: hkavakli@ku.edu.tr

such as *Period* (*Per*) and *Cryptochrome* (*Cry*). The PER, CRY, and casein kinase Iε/Δ (CKIε/Δ) form a trimeric complex and then translocate into the nucleus, where they inhibit BMAL1/CLOCK−driven transcription[6]. After phosphorylation, CRYs are ubiquitinated by E3 ubiquitin ligases (such as FBXL3 and FBXL21), which direct them for proteasomal degradation. CRY's stability in the cytosol and nucleus is controlled by an antagonistic interaction between FBXL3 and FBXL21[7,8].

Retinoic acid receptor-related orphan receptors (RORs) and REV-ERB, encoded by a nuclear receptor subfamily 1 group D member 1 (*NR1D1*), form a second feedback loop. They control the transcription of the *Bmal1* and, in turn, regulate the molecular clock[9]. Recent studies indicated that 50% of detected metabolites in the liver and 43% of transcripts in at least one tissue are under the control of the circadian clock[10,11]. Although circadian clock disruption is related to various cancer types, the loss of *Cry1* and *Cry2* caused an unexpected effect on cancer and did not aggravate the radiation-induced tumor growth and mortality[12]. Furthermore, in the background of cancer-prone *p53*[−/−] mice, the absence of *Cry*s (*Cry1*[−/−] *Cry2*[−/−]) extended the median lifespan 1.5 fold compared to control littermates[13]. These findings suggested that small molecules destabilizing the CRYs can be used as anticancer agents in some cancer types.

Many molecules that affect various features of circadian rhythm have been identified[14–17]. These studies used a high-throughput screening assay to determine the phenotypic changes in the circadian rhythm of reporter cells. A small molecule, KL001, increased the stability of CRYs and suppressed gluconeogenesis[16]. Additionally, the stabilization of CRYs with this molecule results in lengthening of the circadian period at the cellular level[18]. Other studies with KL001 derivatives revealed that despite stabilizing the CRYs, they shorten the circadian period length[18]. Isoform selective KL101 and TH301 molecules stabilizing CRY1 and CRY2, respectively, were shown to regulate the differentiation of brown adipose tissue[19]. ROR agonist, nobiletin, improved the amplitude in mice with metabolic syndrome[20]. Alternatively, a structure-based drug design approach was utilized to design small molecules by using the available crystal structures of core clock proteins[21–23]. For example, a molecule (CLK8) identified from virtual screening was shown to enhance the amplitude of the circadian rhythm by reducing the nuclear levels of CLOCK[15].

In this study, we aimed to find a CRY-binding small molecule using a structure-based approach by targeting the primary pocket (known as the FAD-binding pocket in photolyase), which is critical for the FBXL3 interaction. To this end, a library of ~2 million small molecules was screened using virtual screen methods. We then used several biochemical and molecular tools to find molecules targeting CRY1[24,25]. Finally, we discovered a small molecule (M47) that binds to CRY1, destabilizes it, and lengthens the period of the circadian rhythm at the cellular level. In mice, M47 destabilizes CRY1 in the liver. M47 enhances apoptosis in a Ras-transformed *p53*-null mouse skin fibroblast cell line, which suggests that the molecule makes the cells in *p53*[−/−] mice more sensitive to apoptotic signals and, in turn, may increase their life span. We, therefore, administered M47 to cancer-prone *p53*[−/−] mice to determine its effect on their lifespan. Our results indicated that M47-treated *p53*[−/−] mice had ~25% increased lifespan compared to mock treat control animals. The mild toxicity profile, pharmacokinetic and pharmacodynamics properties of the M47 suggested that it may be used to improve the effectiveness of cancer treatment related to *p53* mutations and other types of diseases related to CRY1.

## Results

### Structure-based small molecule design

Cryptochromes (CRYs) have two distinct domains called PHR and extended C-terminal domains[26]. PHR domain is further divided into two subdomains called primary (known to be important for FAD binding in photolyase) and secondary (important for the binding of the secondary pigment e.g., MTHF in photolyase) pockets, which are critical for protein-protein interactions. For example, the primary pocket is critical for the interaction with FBXL3, an important E3 ubiquitin ligase for the degradation of CRY1[7,8,23]. In this study, the mouse CRY1 (mCRY1) crystal structure (PDB ID: 4KOR) was used for in silico screening by targeting its primary pocket of the PHR domain. To obtain the receptor (CRY1) at the physiological conditions, the molecular dynamics (MD) simulation was performed on the homology model of CRY1. The structure of mCRY1 was solvated in a rectangular box and neutralized with counterions. Subsequently, the system was minimized, heated up to physiological temperature (310 K), and simulated for 50 ns. The equilibrated primary pocket of the CRY1 structure has been used as the receptor for the docking simulations. A freely available library from Ambinter (Orleans-France) containing 8 million small molecules with the non-identified function was used as ligands. The library was initially filtered according to "Lipinski's Rule of Five"[27] and the remaining ~2 million small molecules were screened against the primary pocket of CRY1 using AutoDock Vina (v.1.2). The top 200 compounds were selected for further studies based on their binding energies (range from −9 to −13.5 kcal/mol).

### Identification of the molecules affecting the half-life of the CRY

The toxicity of selected molecules was assessed by MTT assay using human osteosarcoma (U2OS *Bmal1*-d*Luc* where the destabilized form of *Luciferase* (*Luc*) gene fused with *Bmal1* promoter) cells at concentrations of 20, 10, and 2.5 μM. Molecules with relative cell viability less than 90% at 2.5 μM were excluded from further studies. Molecules changing the stability of CRYs were shown to affect the period length of circadian rhythm[16,18]. We treated U2OS *Bmal1*-d*Luc* cells, which let us monitor circadian changes, with the nontoxic small molecules and monitored their bioluminescence to identify molecules changing the period. We identified a set of molecules affecting the circadian rhythm of U2OS *Bmal1*-d*Luc* cells either by lengthening or shortening the period by 1 h or more[24,25]. We reasoned that the molecule possibly changes the half-life of CRY1 and causes the period changes. Therefore, the effects of these molecules on the half-life of CRY1 were determined using CRY1::LUC degradation assay where CRY1 was fused with LUC at the C-terminal as previously described in[16]. Among CRY1 modulating molecules, M47 (Fig. 1A) showed the best dose-responsive effect and was selected for further characterization.

### Characterization of M47

Since we are mostly using luciferase-based reporter assays throughout the study, we tested the effect of M47 on the half-life of dLUC and then its toxicity on U2OS cells in a dose-dependent manner. M47 did not affect the stability of dLUC and show any toxic effect on the U2OS cell line (Fig. S1A, B). We also tested the toxicity of the M47 on primary mouse skin fibroblast cells, where M47 had no toxic effect (Fig. S1C). We then determined the effect of M47 on the circadian rhythm using U2OS *Bmal1*-d*Luc* cells and NIH 3T3 *Bmal1*-d*Luc* cells (kindly provided by Prof John Hogenesch). M47 significantly lengthened the period in a dose-dependent manner in U2OS *Bmal1*-d*Luc* cells (Fig. 1B). M47 treated, which has no toxic effect, NIH 3T3 *Bmal1*-d*Luc* cells exhibited a similar phenotype as well (Fig. S1D, E). However, the effect of M47 was smaller on the period length of NIH 3T3 compared to U2OS. This may be due to the mouse cell line having different genetic backgrounds than the human U2OS cell line. We observed that M47 caused not only period lengthening in these cell lines, but also a dose-dependent decrease in the amplitude of rhythms. A decrease in amplitude is consistent with previous studies where either reduction or deletion of the *CRY1* genes results in low-amplitude rhythms or even arrhythmicity in different cells including U2OS and mutant mice[18,28–32]. This may be due to the fact that *Cry1* is essential for the generation of cell-autonomous circadian clock function[33]. To eliminate the possibility that M47 might affect the other proteins and, in turn, circadian rhythmicity we tested the effect of M47 in *CRY1/CRY2* double knockout

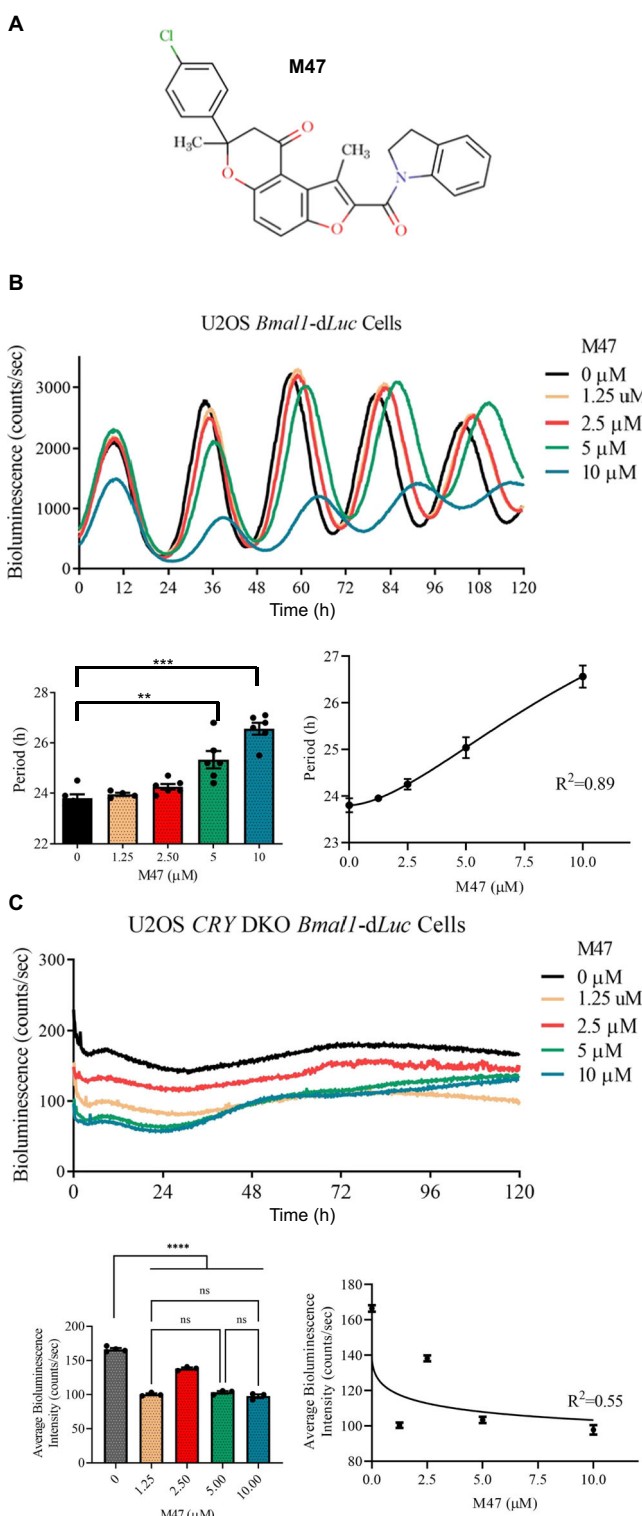

**Fig. 1 | Dose-dependent effect of M47 on the circadian rhythm of U2OS cells.**
**A** Structure of M47. **B** The representative figure for the effect of M47 on the bioluminescence rhythm of U2OS cells stably expressing *Bmal1*-d*Luc*. M47 lengthened the circadian rhythm dose-dependently (Data represent the mean ± SEM, $n = 4$ with duplicates ***$p = 0.0002$, *****$p < 0.0001$, versus DMSO control by one-way ANOVA with Dunnet's post hoc test). **C** Effect of M47 on the *CRY1/CRY2* double knockout (DKO) U2OS cell line (Data represent the mean ± SEM, $n = 3$ with duplicates ns=not significant versus DMSO control by one-way ANOVA with Dunnet's multiple comparison test; ****$p < 0.001$ versus M47 treated cells by one-way ANOVA with Dunnet's multiple comparison test).

(DKO) U2OS *Bmal1*-d*Luc* (Fig. 1C). Results indicated that although all doses of M47 lowered the overall bioluminescence in the absence of the CRYs, the observed effect is not dose-dependent, suggesting that the effect of M47 was through CRY (Fig. 1C).

We next determined the effect of the M47 on the half-life of CRYs by measuring the decay rate of CRY1::LUC and CRY2::LUC as described in[16]. To this end, HEK 293 T cells (kindly provided by Prof Aziz Sancar Group) were transfected with pcDNA-*Cry-Luc* plasmids, treated with M47 doses, then, treated with cycloheximide to inhibit protein translation. Notably, M47 reduces the half-life of CRY1 in a dose-dependent manner while there was no effect on the half-life of CRY2 (Fig. 2A, B). To verify the CRY1 dependency of M47, we generated *CRY1* knockout U2OS *Bmal1*-d*Luc* cells by utilizing CRISPR/Cas9 technology (Fig. S1F). Knocking out the *CRY1* in this cell line resulted in a shorter period compared to wild-type control as in agreement with previously published data[34] (Fig. 2C). When *CRY1* knockout U2OS *Bmal1*-d*Luc* cells were treated with the M47, no change was observed in the period length of the circadian rhythm (Fig. 2C). We also used *CRY2* knockout U2OS *Bmal1*-d*Luc* cell (Fig. S1F) to assess the selectivity of M47 on CRY1. In agreement with CRY degradation assay, M47 treatment increased the period length of *CRY2* knockout U2OS *Bmal1*-d*Luc* cells in a dose-dependent manner (Fig. 2D). Collectively, CRY degradation and circadian phenotypes of *CRY1* knockout, *CRY2* knockout and wild type U2OS *Bmal1*-d*Luc* cells showed that the M47 selectively binds to the CRY1 and exerts its effect through CRY1.

The docking pose of M47 showed that it binds to the "FAD-binding" region of the CRY1 by interacting with Arg293, Trp292, Ser396, and Trp399 amino acid residues (Fig. 2E). Indoline groups of M47 interact with Trp292 through a strong pi-alkyl interaction and with Arg293 through van der Waals interaction (Fig. 2E). Additionally, benzofuran moiety of the M47 interacts with indole groups of Trp292 and Trp399 (Fig. 2E) through pi type interactions. We hypothesized that the replacement of these residues in CRY1 should eliminate the binding of M47 to CRY1. To test this, Arg293 and Trp399 of CRY1 were replaced with Ala and Leu by site-directed mutagenesis, respectively. Before measuring the effect of M47 on this CRY1 mutant, we confirmed CRY1-R293A-W399L mutant retained its repressor activity with *Per1*-d*Luc* assay using BMAL1/CLOCK in the presence of the mutant and wild-type CRY1 (Fig. S2A). We then measured the half-life of CRY1-R293A-W399L::LUC in the presence of M47. The molecule did not affect the half-life of CRY1- R293A-W399L compared to DMSO control (Fig. 2F). This suggested that M47 binds the computationally predicted amino acids in the "FAD-binding" region of CRY1.

We hypothesized that M47 binds to the PHR domain and increases the degradation rate of CRY1 by increasing its ubiquitination level[35–37]. To test this HEK293T cells were transfected with pcDNA-*Cry1*-His-Myc, pBIND-*Fbxl3*-GAL4, and pUb-HA plasmids in the presence and absence of M47. To determine the ubiquitination level of CRY1, proteasomal degradation was blocked by MG132. Proteins were isolated with anti-MYC resin and ubiquitination levels were determined by Western blot. The analysis of the ubiquitination assay revealed that CRY1 is hyper-ubiquitinated when treated with M47 compared to DMSO control (Fig. 2G). Some small molecules are defined as "molecular glues" that enhance protein-protein interactions via binding the interface through different mechanisms of action between various types of proteins[38]. Ubiquitin ligases and their substrate interfaces are also within targets of so-called molecular glues, therefore facilitating these substrates' poly-ubiquitination and proteasomal degradation[39–41]. Therefore, M47 might act as a glue molecule between CRY1 and FBXL3 and in turn, stabilize the interaction between them and may cause hyper ubiquitination. However, the exact mechanism in terms of the interactions between them can be obtained from the crystal structure of the CRY1-FBXL3-M47.

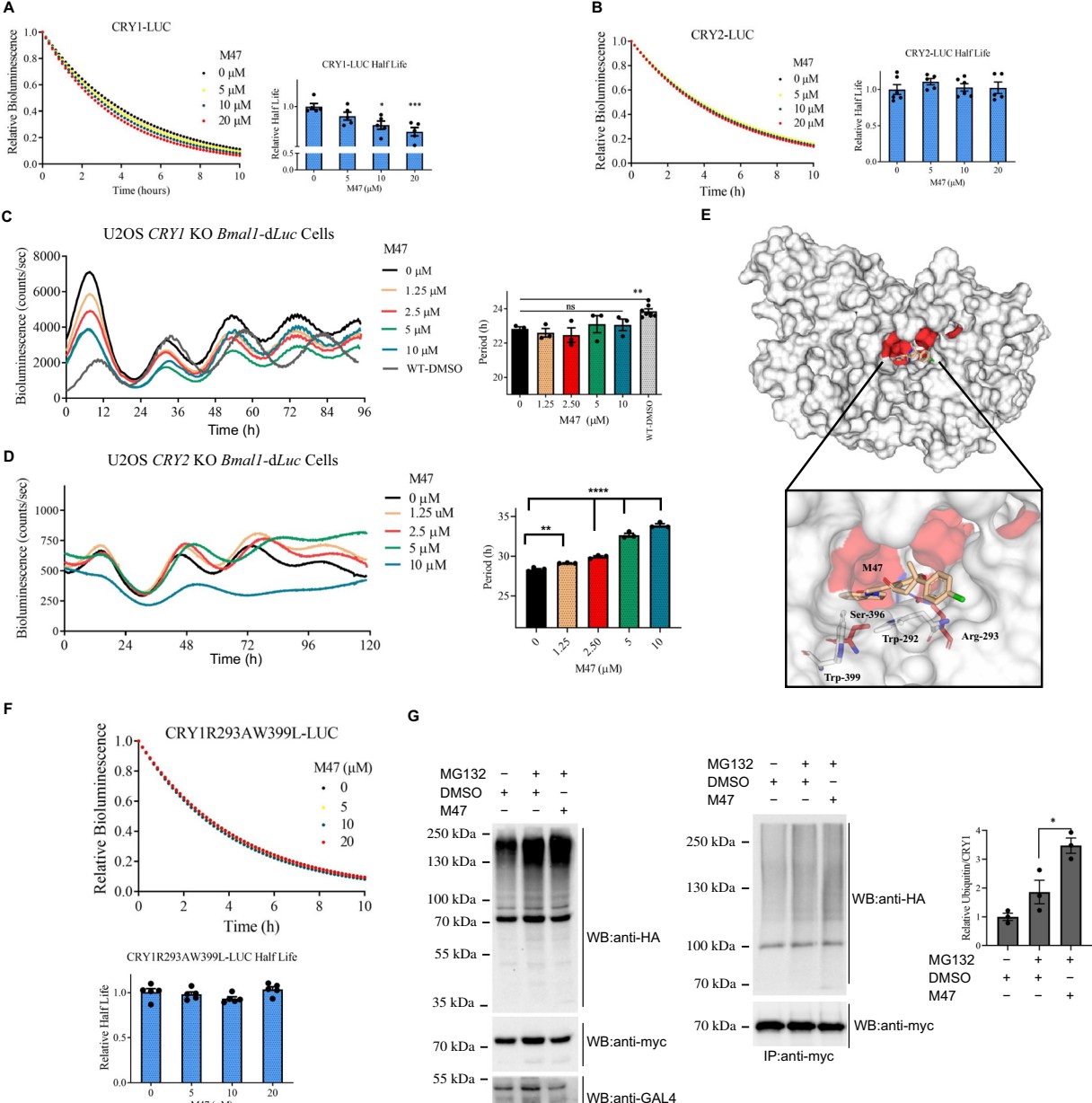

**Fig. 2 | The effect of M47 on the half-life of CRYs. A** M47 increased the degradation rate of CRY1::LUC dose-dependently. Normalized half-life is shown with ± SEM (*n* = 4 with triplicates). *\*p* = 0.01, *\*\*\*p* = 0.0008 versus DMSO treated cells with one-way ANOVA with Dunnet's multiple comparison test. **B** M47 did not affect the degradation rate of CRY2::LUC. Normalized half-life is shown with ± SEM (*n* = 5 with triplicates one-way ANOVA with Dunnet's multiple comparison test). **C** The representative figure for the effect of M47 on the bioluminescence rhythm of U2OS *CRY1* knockout (KO) *Bmal1*-d*Luc*. M47 did not affect on the circadian rhythm of the *CRY1* knockout U2OS *Bmal1*-d*Luc* in dose-dependently (Data represent the mean ± SEM, *n* = 3 with duplicates *\*\*p* = 0.002, wild-type U2OS *Bmal1*-d*Luc* versus CRY1 KO U2OS *Bmal1*-d*Luc* by two-tailed student's *t*-test; ns = statistically not significant one-way ANOVA in *CRY1* KO U2OS *Bmal1*-d*Luc* DMSO vs M47 treatment).

**D** The representative figure for the effect of M47 on the bioluminescence rhythm of *CRY2* knockout U2OS *Bmal1*-d*Luc*. M47 increased the period length of the circadian rhythm in U2OS *CRY2* KO *Bmal1*-d*Luc* in dose-dependently (Data represent the mean ± SEM, *n* = 3 with duplicates *\*\*p* = 0.0072 and *\*\*\*\*p* < 0.0001 versus DMSO control by one-way ANOVA with Dunnet's post hoc test). **E** Binding pose of M47 on the simulated CRY1 structure predicted by AutodockVina with −11.2 kcal/mol binding energy. Protein structure is shown in surface. FBXL3 binding residues are colored as red. M47 interacting residues are shown in sticks. **F** M47 did not affect the half-life of mutant CRY1R293AW399L-LUC degradation Normalized half-life is shown with ± SEM (*n* = 4 with triplicates with one-way ANOVA test). **G** The ubiquitination of the CRY1 in the presence of M47. (Data represent the mean ± SEM, *n* = 3; *\*p* = 0.0297 versus DMSO control with *t*-test with two-tailed).

All these results showed that M47 binds to the primary pocket in the PHR domain and specifically destabilizes CRY1 by enhancing its ubiquitination.

### Investigation of M47 and CRY1 interaction

The biotinylated-M47 (bM47) (Fig. S2B) was used to show the physical interaction with CRY1. Plasmids containing His-Myc tagged *Cry1* and *Cry2* were transfected into HEK293T cells to see if bM47 binds

CRYs. CRY1-HIS-MYC and CRY2-HIS-MYC were pulled down using bM47 in the presence and absence of the competitor (free M47). M47 was shown to specifically bind to CRY1 but not CRY2, and this interaction highly reduced in the presence of the competitor (Fig. 3A). A pull-down experiment between bM47 and CRY1-T5 (the lack of 100 amino acids from the C-terminal end) verified that M47 interacts with the PHR domain of the CRY1 (Fig. 3B). The effect of bM47 on the circadian rhythm and half-life of CRY1 was tested to

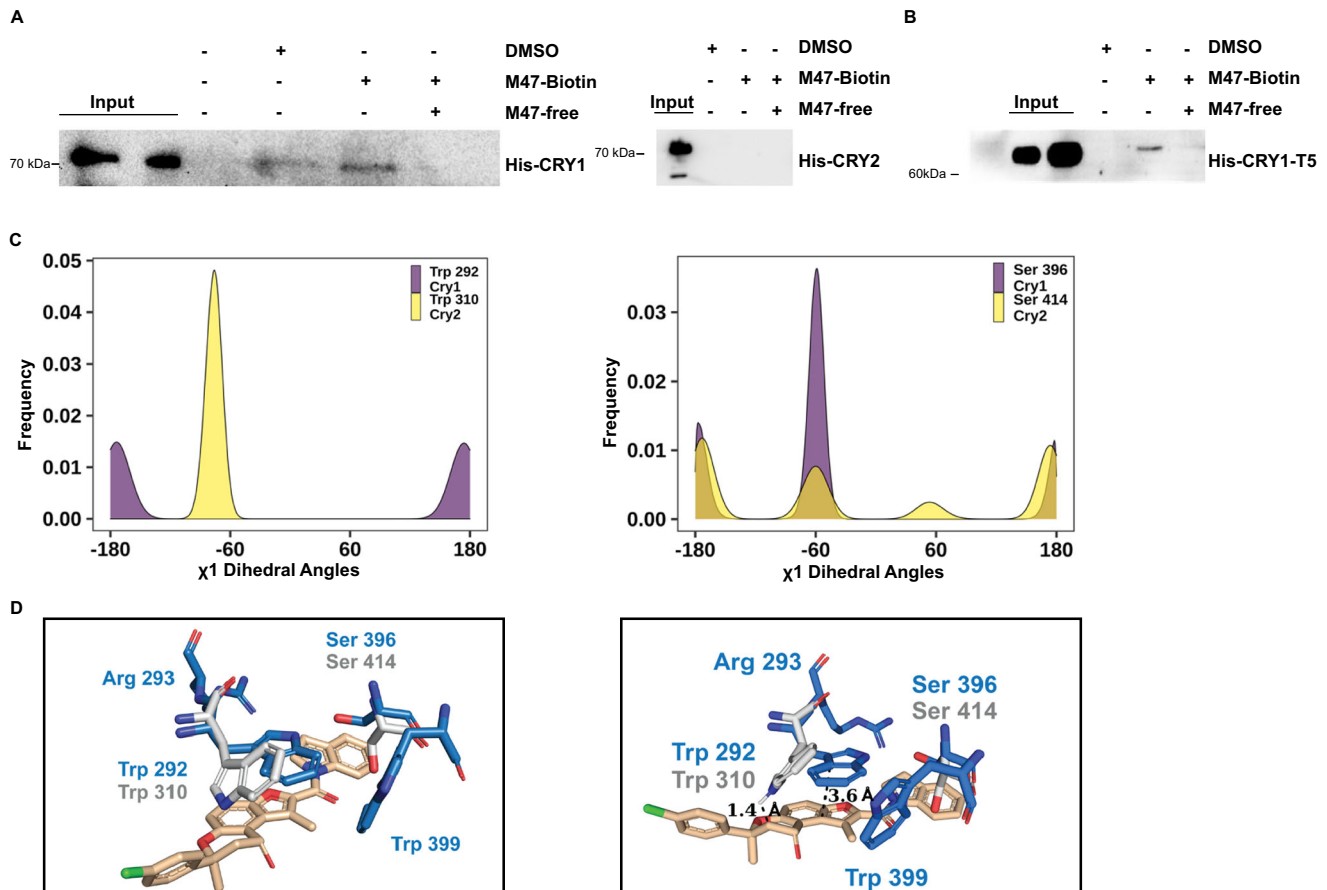

**Fig. 3 | Physical interaction between M47 and CRYs. A**, **B** HEK293T cells transfected with pcDNA4A-*Cry1-His-Myc*, pcDNA4A-*Cry2-His-Myc*, or pcDNA4A-*Cry1-T5-His-Myc* plasmids then lysates were subjected to pull-down assay. Lysates were treated with solvent (DMSO), 50 μM bM47, and 50 μM bM47 with 100 μM M47 (competitor). While M47 binds to the PHR of CRY1 (**A**, **B**) it does not bind to CRY2 (**A**) (*n* = 3). **C** χ1 dihedral angle distribution M47 interacting residues of CRY1 and CRY2 throughout 300 ns MD simulation. **D** Two superimpose images from two different angles of CRY1 and CRY2 structures. M47 binding residues (Trp399, Trp292, Arg293, and Ser396) in CRY1 and corresponding residues in CRY2 (Trp310, Ser414) are shown in sticks.

ensure that biotinylation did not change the molecule's function (Fig. S2C, D). We docked bM47 on the primary packet of CRY1 and found that it has a similar binding mode compared to M47 with a slight shift. We measured the distance between the center of masses M47 and bM47 (excluding the biotin atoms) as 1.6 Å (Fig. S2E). All these experimental and computational studies suggest that M47 binds to the primary pocket of the CRY1. For the LC-MS/MS analysis we used bM47 for the pull-down study using protein extract prepared from HEK293T cell line in the presence and absence of a competitor (M47) as described by Hirota et al.[16] The results are given in Table 1. The bM47 binds to CRY1 and its interaction decreased in the presence of a competitor. The other three identified proteins were determined as sticky proteins from Crapome Database.

To understand why CRY2 was unable to bind M47 we performed 300 ns MD simulation of CRY1-PHR and CRY2-PHR (Fig. S3A). We superimposed the best binding conformation of M47 with CRY1 to the most crowded cluster representative of CRY2. There was a steric clash between M47 and CRY2 due to different orientations of Trp310 and Ser414 (Fig. 3C). To probe the orientation of these amino acids in entire simulations of CRY1 and CRY2, χ1 dihedral angle of Trp292 (Trp310 in CRY2) and Ser396 (Ser414 in CRY2) were calculated. The distribution of these dihedral angles was about −65° conformation for Trp310 in CRY2 while Trp292 in CRY1 was at trans conformation (Fig. 3D). Our χ1 dihedral angle distribution Trp292 of CRY1 is similar to previously published MD results of CRY1, where 5T5X structure was used as initial structure (Fig. S3B–E)[42]. Ser396 of CRY1 exhibited major peaks around

−60° but Ser414 of CRY2 showed multiple alternate conformers at −180°, −60°, 60°, and 180°. Although Ser414 of CRY2 causes a steric clash in the representative frame, χ1 dihedral angle distribution show that it has multiple conformers. Therefore, it may avoid steric clashes with M47. All these results suggest that, due to their different predominant conformations, the amino acids involved in the selectivity of M47 are Trp292 in CRY1 and Trp310 in CRY2.

These results showed that despite CRYs have highly conserved PHR regions, their internal dynamics are different which can regulate their cellular functions and interactions with other proteins and ligands.

### Time-dependent effect of M47 on the U2OS cells

To evaluate the effect of M47 on the endogenous clock proteins and their mRNA levels, synchronized U2OS cells were treated with 10 μM M47. Cells were harvested between 24 and 48 h of post-synchronization at 4 h time intervals. M47 decreased the level of CRY1, at all the time points but 36th h (Fig. 4A). On the other hand, we observed no significant change in CRY2 and CLOCK levels. There was a slight change in the BMAL1 level. This is possibly due to the reduction of CRY1 level might have increased the REV-ERBα, which is a repressor of BMAL1 expression. Increased level of the REV-ERBα might cause the reduction of BMAL1 level. Analysis of mRNA levels of *CRY1* and *CRY2* showed that M47 did not change their overall abundance (Fig. 4B). An increase in clock output genes of the *DBP* and *PER2* mRNA levels also confirmed the decrease in the total repressor capacity of the cells

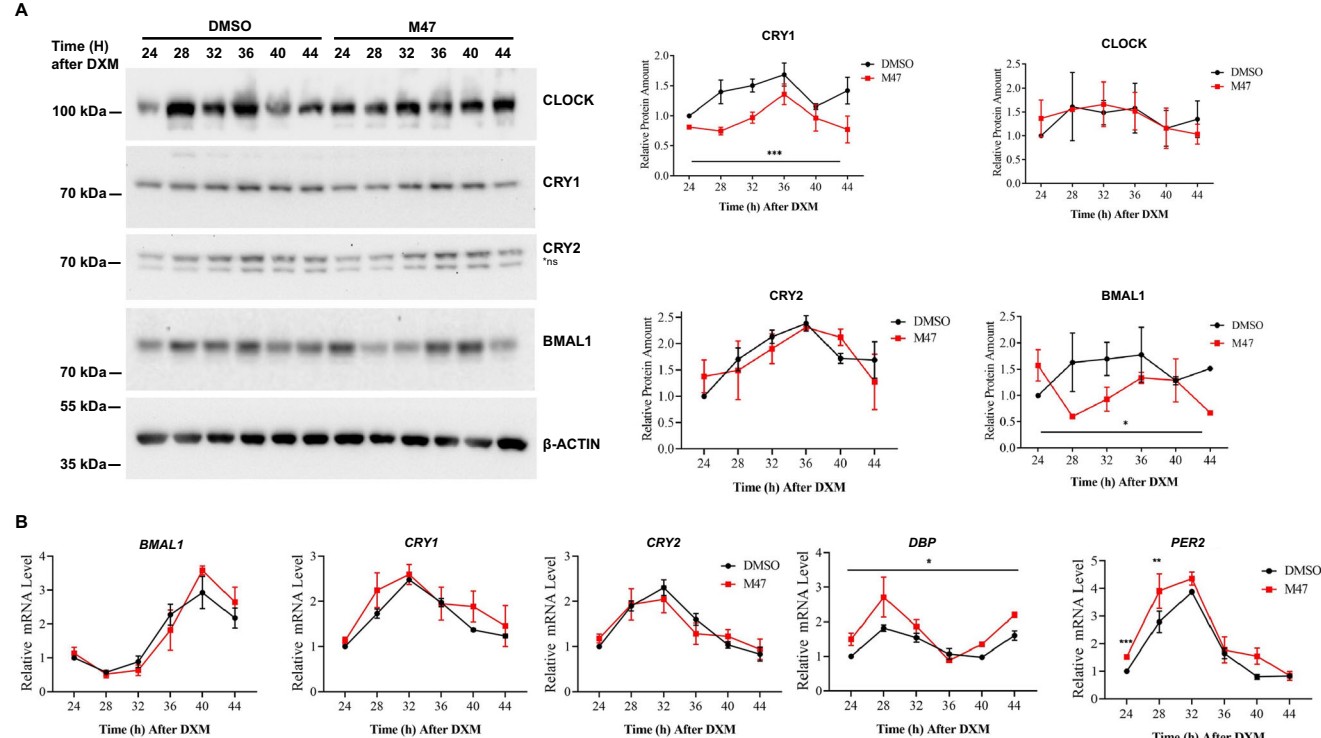

**Fig. 4 | The effect of M47 on core clock genes and proteins in synchronized U2OS cells.** Confluent U2OS cells were synchronized by 2 h treatment with dexamethasone (0.1 μM) and medium replaced with fresh medium containing M47 or solvent (DMSO final 0.5%). Cells were harvested at indicated time points. **A** The protein level in lysed cells was analyzed by the protein immunoblot technique. M47 decreased the protein level of CRY1 between 24th–48th h. The level of proteins was normalized to DMSO at 24th h (mean ± SEM, $n = 3$). (*$p = 0.0286$, ***$p = 0.0001$ versus DMSO control by two-way ANOVA with Dunnet's multiple comparison test)

(**B**) Cells were subjected to a reverse-transcription-quantitative polymerase chain reaction (RT-qPCR). M47 treatment increased overall *Dbp* level and not affect the *Cry1* and *Cry2* levels. At 24 and 28th h M47 increased *Per2* (**$p = 0.0082$, ***$p = 0.0003$ versus DMSO control with *t*-test with two-tailed). The expression level of genes was normalized relative to the expression level of that gene at 24th h. (Data represent the mean ± SEM, $n = 3$ with duplicates). *ns non-specific band in western blot.

treated with M47. Observing the effect of M47 on CRY1 at the protein level but not at the mRNA level confirms the post-translational effect of the molecule. All these results suggest that M47 is a promising molecule to regulate circadian clock machinery through CRY1. To investigate the pharmacological properties and activity of M47 in vivo, it has been subjected to mice studies.

## Toxicity analyses of M47 in mice

We first carried out a single-dose toxicity study (SDT) to assess M47's toxicity in C57BL/6 J mice. Male and female mice were administered intraperitoneally (i.p.) with doses of 5, 50, 300, and 1000 mg/kg. We evaluated the general toxicity in the animals based on body temperature, body weight changes, clinical signs, mortality, water and food consumption, gross findings during the necropsy of death, and evaluation of behavior. The following clinical symptoms were observed from animals administered with 1000 mg/kg of the M47: reduced locomotor activity, hunched posture, dyspnea, excessive hyperthermia, piloerection, hyporeflexia, and transparent cornea. For ethical concerns, animals were sacrificed after receiving this dose, which was thought to be toxic. Other doses (5, 50, and 300 mg/kg) did not result in any death or clinical symptoms. Mild hypothermia (33.5–35.1 °C) was observed in mice within 6 h of receiving the M47 and their body temperatures gradually increased at the dose of 300 mg/kg, compared to control mice (given with the vehicle), where their body temperatures were 36.0–37.0 °C (Fig. 5A). At a dose of 5 mg/kg, the animals' body temperatures were higher than those of the control groups. However, the animals' body temperatures were comparable when 50 mg/kg of M47 was injected (Fig. 5A). We then assessed weight changes in these animals for 15 days. The greatest mean body weight

loss (7.7%) occurred on the first day following 300 mg/kg M47 treatment (Fig. 5B). Animals administered with 5 mg/kg of the M47 exhibit no appreciable weight reduction. All of these findings revealed that SDTs of 5 and 50 mg/kg were well tolerated by mice, while mice exhibited some adverse effects at 300 mg/kg.

The 5-day maximum tolerated dose (MTD) was then determined using 40, 80, or 150 mg/kg of i.p. M47 with repeated doses over 5 days. Among all tested doses, one mortality (1/6; 16.6%) was noted in the 150 mg/kg group on the third day of injection. There were no clinical indications in the animals administered with 40 and 80 mg/kg. Animals administered 150 mg/kg showed hyporeflexia, which was more pronounced during the first three days of therapy and diminished on fourth and fifth day. The animals given 40 mg/kg had body temperatures that were similar to those of the control animals. However, mice treated with 80 and 150 mg/kg had mild hypothermia (33.4–35.8 °C) on the first day. In the following days, there was a progressive rise in body temperature (Fig. 5C). The body weight changes were similar for all dosages, with the exception of the third and fourth injections at the dose of 150 mg/kg, where the highest mean body weight loss (6.3%) was recorded on the third day of injection (Fig. 5D).

All of these findings suggested that mice tolerated 5 and 80 mg/kg. As a result, we decided to conduct a subacute toxicity test on animals, using 60 mg/kg for 14-days of repeated injections. Neither clinical symptoms nor mortality were observed animals received M47 once a day for two weeks. Repeated injections caused mild hypothermia (33.7–35.4 °C) (Fig. 6A). Although animals had the greatest mean body weight decrease (3.7%) at first day of injection, body weight was recovered in subsequent days (Fig. 6B).

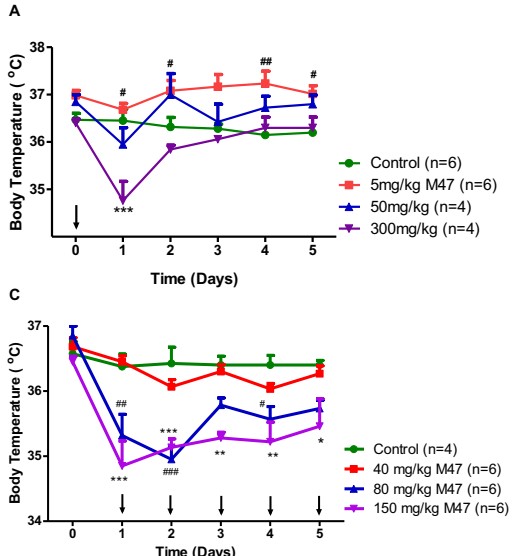

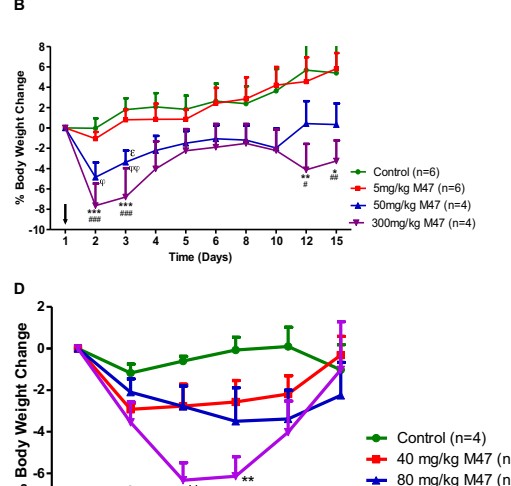

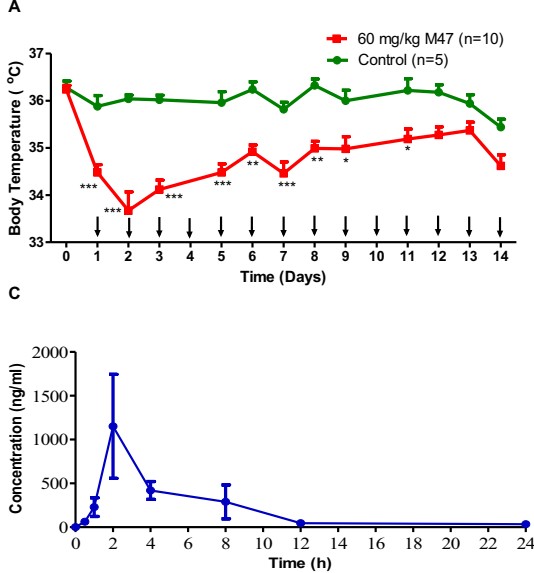

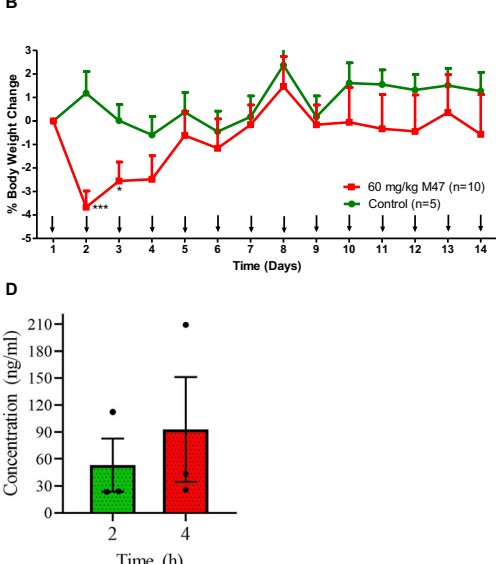

**Fig. 5 | Toxicity of M47 in mice. A** Body temperatures in C57BL/6 J mice ($n > 4$) treated with single i.p. doses of 5 mg/kg, 50 mg/kg, 300 mg/kg M47 or vehicle during 15-day observation period. Body temperatures (°C) were expressed as mean ± SEM. "↓" indicates the treatment days of M47. ***$p < 0.001$, control vs 300 mg/kg M47; #$p < 0.05$, ##$p < 0.01$ control vs 5 mg/kg M47 (Two-way ANOVA with Bonferroni post hoc test). **B** Body weight changes (%) in C57BL/6 J mice ($n > 4$) treated with single i.p. doses of 5 mg/kg, 50 mg/kg, 300 mg/kg M47 or vehicle during 15-day observation period. Body weight changes (%) were expressed as mean ± SEM. "↓" indicates the treatment days of M47. *$p < 0.05$, **$p < 0.01$, control vs 300 mg/kg M47; ###$p < 0.001$ 5 mg/kg vs 300 mg/kg M47; φ$p < 0.05$, φφ$p < 0.01$ control vs 50 mg/kg; ε$p < 0.05$, 5 mg/kg vs 50 mg/kg M47 (Two-way ANOVA with

Bonferroni post hoc test). **C** Body temperatures in C57BL/6 J mice ($n > 4$) treated with i.p. doses of 40 mg/kg, 80 mg/kg, 150 mg/kg M47 or vehicle for 5 days. Body temperatures (°C) were expressed as mean ± SEM. "↓" indicates the treatment days of M47. *$p < 0.05$, **$p < 0.01$, ***$p < 0.001$, control vs 150 mg/kg M47; #$p < 0.05$, ##$p < 0.01$, ###$p < 0.001$ control vs 80 mg/kg M47 (Two-way ANOVA with Bonferroni post hoc test). **D** Body weight changes (%) in C57BL/6 J mice ($n > 4$) treated with i.p. doses of 40 mg/kg, 80 mg/kg, 150 mg/kg M47 or vehicle for 5 days. Body weight changes (%) were expressed as mean ± SEM. "↓" indicates the treatment days of M47. **$p < 0.01$ control vs 150 mg/kg M47 (Two-way ANOVA with Bonferroni post hoc test). All the sample sizes ($n$) are given in the figures.

**Fig. 6 | Subacute toxicity and pharmacokinetics of M47. A** Body temperatures in C57BL/6 J mice ($n > 5$) treated with i.p. dose 60 mg/kg M47 or vehicle for 14 days. Body temperatures (°C) were expressed as mean ± SEM. "↓" indicates the treatment days of M47. *$p < 0.05$, **$p < 0.01$, ***$p < 0.001$, control vs 60 mg/kg M47 (Two-way ANOVA with Bonferroni post hoc test). **B** Body weight changes (%) in C57BL/6 J mice ($n > 5$) treated with i.p. dose of 60 mg/kg M47 or vehicle for 14 days. Body weight changes (%) were expressed as mean ± SEM. "↓" indicates

the treatment days of M47. *$p < 0.05$, ***$p < 0.001$, control vs 60 mg/kg M47 (Two-way ANOVA with Bonferroni post hoc test). **C** Mean plasma concentration-time curve of M47 administered at 100 mg/kg i.p. to C57BL/6 J mice. Data were expressed as mean ± SEM ($n = 4$ per time point). **D** M47 levels in brain tissue at 2 h ($n = 3$) and 4 h ($n = 3$) following M47 administration at a dose of 100 mg/kg i.p. to C57BL/6 J mice. Data were expressed as mean ± SEM. All the sample sizes ($n$) are given in the figures.

**Table 1 | Identification of potential M47 binding proteins using LC-MS/MS**

| | Experiment 1 | | | Experiment 2 | | | Experiment 3 | | | Fold Change | Crapome Results | | |
|---|---|---|---|---|---|---|---|---|---|---|---|---|---|
| | Control (DMSO) | Biotin-M47 | Biotin-M47 / Competitor | Control (DMSO) | Biotin-M47 | Biotin-M47 / Competitor | Control (DMSO) | Biotin-M47 | Biotin-M47 / Competitor | | Num of Expt. (found/ total) | Average SC | Max SC |
| CRY1 (Cryptochrome 1) | 0 | 6 | 6 | 0 | 6 | 0 | 0 | 8 | 4 | 5.0 | /716 | 0 | 0 |
| RPS15A (40S ribosomal protein S15a) | 0 | 12 | 0 | 0 | 8 | 4 | 0 | 12 | 6 | 3.2 | 401 / 716 | 5.7 | 34 |
| HSP90AA1 (Heat shock protein HSP 90-alpha) | 0 | 28 | 22 | 0 | 24 | 46 | 0 | 0 | 0 | 0.8 | 565 / 716 | 33.7 | 343 |
| YWHAE (14-3-3 protein epsilon) | 0 | 8 | 4 | 0 | 14 | 2 | 0 | 8 | 2 | 3.8 | 447 / 716 | 10 | 81 |

Streptavidin conjugated agarose is used for pull-down assay form HEK 293 T cells incubated with biotinylated M47 in the absence (DMSO) and presences of competitor (M47). AP-MS results were analyzed as follows: All proteins that have MS spectra greater than 5 in DMSO (control) and less than 5 MS spectra in pull-down with biotinylated M47 in each of three replicates were filtered. Surviving protein list was submitted to Crapome Database to identify nonspecific sticky interactions. AP-MS: Affinity purification-mass spectrometry.
SC spectral count.

**Table 2 | Plasma pharmacokinetic parameters of M47 (100 mg/kg, single dose, i.p.)**

| Parameters | Values |
|---|---|
| $C_{max}$ (ng/ml) | $1150.52 \pm 506.26$ |
| $T_{max}$ (h) | 2–4 |
| $AUC_{0-24}$ (ng.h/ml) | $4921 \pm 3069.09$ |
| $AUC_{0-\infty}$ (ng.h/ml) | $5194 \pm 3134.00$ |
| $k_{el}$ (1/h) | $0.127 \pm 0.01$ |
| $t_{1/2}$ (h) | $5.5 \pm 0.33$ |

## Pharmacokinetic profile of M47 in mice

The pharmacokinetic parameters and the mean plasma concentration-time curve of M47 were given in Table 2 and Fig. 6C, respectively. In this study, female C57BL/6 J mice were administered 100 mg/kg of M47 single dose (i.p.). The presence of M47 in the plasma was identified at 0.5 h, peaked at 2–4 h, and remained quantifiable at 24 h. M47 levels in the brain were only assessed at the 2nd and 4th h ($n = 3$) and it was detectable at both time points (Fig. 6D). All of these findings indicated that the M47 passes the blood-brain barrier and has a half-life of 6 h in vivo.

## The effects of M47 in mice liver

All in vitro experiments indicated that M47 binds and shortens CRY1's half-life. M47 was i.p. injected into a mouse at doses of 25 mg/kg and 50 mg/kg to evaluate its in vivo effect. Livers were collected for further investigation after the mice were sacrificed at 6 h following the injection. The CRY1 level was slightly decreased in cell lysate of mouse liver (Fig. 7A). The level of CRY1 was also measured in both the nucleus and the cytosol of the mice liver because of its differential stability in subcellular fractions[7,8]. To assess the purity of the cellular fractionations, proteins known to be selectively localized in the nucleus (Histone-H3) and the cytoplasm (Tubulin) were blotted as controls (Fig. S4). The cytosolic levels of CRY1 were not statistically different from controls in the mice liver after M47 treatment (Fig. 7B). CRY1 level in the nucleus, on the other hand, was significantly lower in a dose-dependent manner compared to controls (Fig. 7C). *Per2* mRNA levels were assessed using qPCR to examine the effect of M47 on clock-dependent transcription. The expression level of *Per2* increased in mice given M47 as consistent with the nucleus's lower amount of CRY1, which is in agreement with a previously published study[31] (Fig. 7D). All these suggested that M47 reduces the half-life of CRY1 in mouse liver.

## Investigation of the effect of M47 in *p53*⁻/⁻ mouse skin fibroblast cells in the presence of oxaliplatin

A previous study with mutant mouse skin fibroblast (MSF; *Cry1*⁻/⁻ *Cry2*⁻/⁻*p53*⁻/⁻ and *p53*⁻/⁻) indicates that apoptosis is enhanced when *Cry* is deleted on the *p53*-null background[13]. They showed this apoptosis is mediated by UV or UV mimetic agents e.g oxaliplatin. Since M47 destabilizes CRY1, we hypothesized that treatment of *p53*-null MSF cell line with M47 and oxaliplatin may increase the apoptosis. To this end, first the *p53*-null MSF cell line was treated with M47 for 24 h and then followed by oxaliplatin treatment for 16 h. We then measured the apoptosis by probing for PARP cleavage. Since the *Cry* mutation makes *p53*-null cells more susceptible to apoptosis, oxaliplatin was chosen as the preferable chemotherapy agent[43]. We showed that M47 significantly increased the sensitivity of *p53*-null MSF cell line to apoptosis (Fig. 8A). In the same setup, we checked the level of CRY1 and verified that the CRY1 levels were reduced. We further show apoptosis in these cell lines by measuring Caspase-3 activity. The *p53*-null cells treated with M47 exhibited increased Caspase-3 activity under the increasing concentration of the oxaliplatin compared to cells treated with DMSO (Fig. 8B). On the other hand, wild-type MSF cells had comparable Caspase-3 activity under different concentration

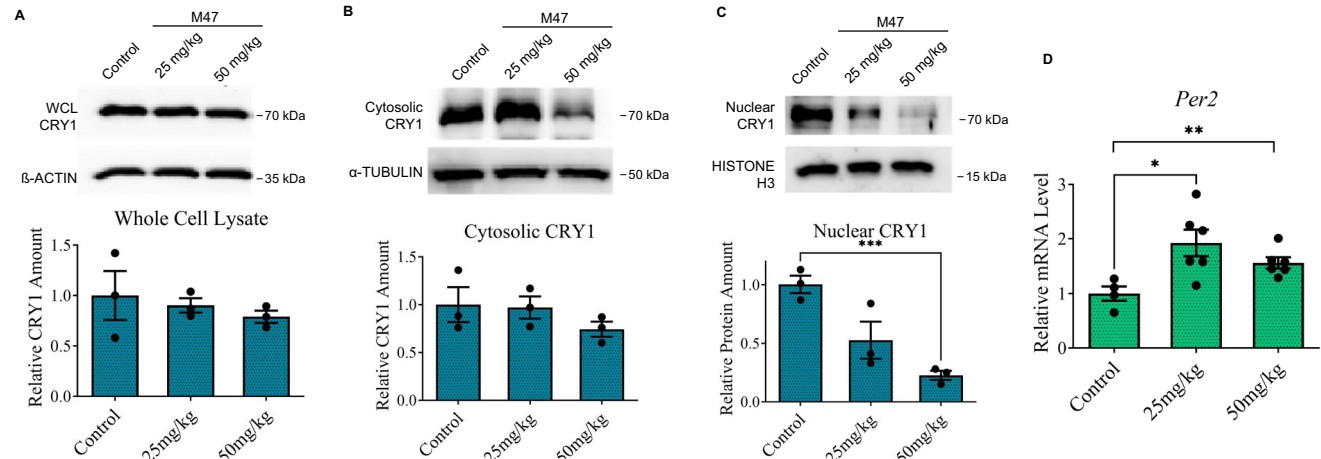

**Fig. 7 | The effect of M47 on CRY1 and *Per2* expression levels in mice liver.** M47 (25 mg/kg or 50 mg/kg) or vehicle i.p. administered to C57BL/6J mice. 6 h after the treatment mice were sacrificed (*n* = 3). While M47 slightly decreased the CRY1 levels in whole cell lysate (WCL) and cytosolic fraction, decreased the nuclear CRY1 level significantly. Effect of M47 in (**A**) WCL (**B**) cytosolic fraction, and (**C**) nuclear fraction of mice liver. M47 treatment caused the reduction of the CRY1 level in the nucleus (Data represent the mean ± SEM, *n* = 3, \*\**p* = 0.0008 versus control by *t*-test two-tailed). **D** Liver samples subjected to RT-PCR. M47 treatment increased the mRNA level of *Per2*. (Data represent the mean ± SEM, *n* = 3 (with duplicates in RT-PCR) \**p* = 0.0211 and \*\**p* = 0.0082 versus control by *t*-test two-tailed).

of oxaliplatin. All these results suggest that M47 enhances the apoptosis in *p53*-null MSF cell lines.

### Effect of M47 on the lifespan of *p53*$^{-/-}$ mice

All our results showed that M47 selectively reduces the half-life of CRY1 and is well tolerated with a good pharmacokinetic profile in vivo and enhances apoptosis. Studies showed that male *p53*$^{-/-}$ mice mostly develop lymphomas and lymphoid sarcomas with an average of 20 weeks in C57BL/6J background[44,45]. Additionally, as previously shown, deletion of *Cry* genes in *p53*$^{-/-}$ in C57BL/6J background enhances the lifespan of the animals by protecting animals from cancer death[13]. A recent transcriptomic study revealed that differential regulation of nuclear factor kappa B (NF-κB) regulation may increase the survival rate[46]. We wished to test M47 on *p53*$^{-/-}$ animals to see whether it increases the lifespan in cancer-prone *p53*$^{-/-}$ C57BL/6J mice. To test that 50 mg/kg/day M47 was i.p. administered to male *p53*$^{-/-}$ mice for 5 days per week at ZT17, for 7 consecutive weeks (*n* = 9) along with vehicle-treated control animals (*n* = 7). It was known that the CRY1 level peaks at ZT22[47]. To minimize the level CRY1 abundance M47 was administered at ZT17 to reduce total CRY1 abundance in tissues considering the half-life of M47 as 6 h. As seen in Fig. 8C, the age-adjusted survival in M47-treated *p53*$^{-/-}$ mice was significantly higher than that of vehicle-treated *p53*$^{-/-}$ mice. The median lifespan of M47-treated *p53*$^{-/-}$ mice was 29.5 weeks while the median lifespan of vehicle-treated *p53*$^{-/-}$ mice was 24 weeks. Further work is needed to provide mechanistic explanations for the reduction of cancer risk in *p53*$^{-/-}$ mice by M47 treatment.

### Discussion

Since circadian rhythm greatly affect human health, it's crucial to prevent circadian disturbances and associated health implications[48]. The development of methods that conditionally and selectively regulate circadian rhythm in controllable way is necessary[49]. For several reasons, intervention programs including genetic alterations or modifications to a patient's regular behavior are not preferable. Using a drug that specifically targets the molecular clock would be a more promising therapeutic approach. Currently, a growing number of groups are investigating molecules that modulate environmental cues and the molecular clock. Therefore, molecules that modulate core clock protein activities and their modifiers were identified using high-throughput screening such as casein kinase Iε (CK1ε), glycogen

synthase kinase 3β (GSK3β), AMP-activated protein kinase (AMPK), and SIRT1[50–52], CRY[16,18,53], REV-ERBs[17,54], and RORs[33,55].

In this study, a CRY1-binding molecule, M47 that decreases the half-life and activity of CRY1, was discovered using the structure-based drug design method. Our results showed that M47 binds to CRY1 but not CRY2. M47 decreases the total abundance of CRY1 in both U2OS cells and mouse liver cells. When CRY1::LUC overexpressed in HEK293T cells, M47 shortens the half-life of CRY1. In synchronized U2OS cells, M47 decreases CRY1 level at certain time points and increases the transcription of clock-controlled genes, e.g., *PER2* and *DBP*. Docking simulations revealed that M47 binds to the PHR domain of CRY1, which is critical for generating the circadian rhythm of the cells[33]. Docking pose and mutational analysis show that M47 binds to Trp399 and Arg293 in the primary pocket. Despite a decrease in the level of CRY1, M47 lengthens the period and dampens the amplitude in U2OS *Bmal1*-dLuc cells. This is possibly due to the potent repressor activity of the CRY1 as shown previously[33]. Further studies using cell-based genetic complementation reveal that instead of the stability and the amount of CRY, its subcellular location and interactions with other core clock proteins influence the period length[5,15,56–58]. Although period-shortening molecules were expected to be CRY destabilizers, GO044, GO200, and GO211 (period-shortenings) were shown to increase the stability of CRY[18]. This provides compelling evidence for the complex relationship between CRY half-life and circadian period length.

M47 binds to the primary pocket of CRY1 by interacting with Arg293, Trp292, and Trp399 amino acids residues. Although these residues are conserved in CRY2, their conformational positions are different in both proteins. Due to the steric hindrance mediated by Trp310 in CRY2, M47 was particularly able to bind to CRY1 but not CRY2 suggesting distinct dynamics between them. It may be possible to use the variations between CRYs to design small molecules that selectively control the activity of each protein. This would help us understand the functions of CRY1 and CRY2 in the clockwork mechanism.

M47, which has a half-life of around 6 h in plasma, is well-tolerated by animals when administered with single and repeated subacute doses. M47 was shown to cross the blood-brain barrier without changing the hematological parameters. All these results suggest that M47 may be a drug-like molecule. Further in vivo studies indicated that M47 also interacts with CRY1 and reduces its level in the mouse liver.

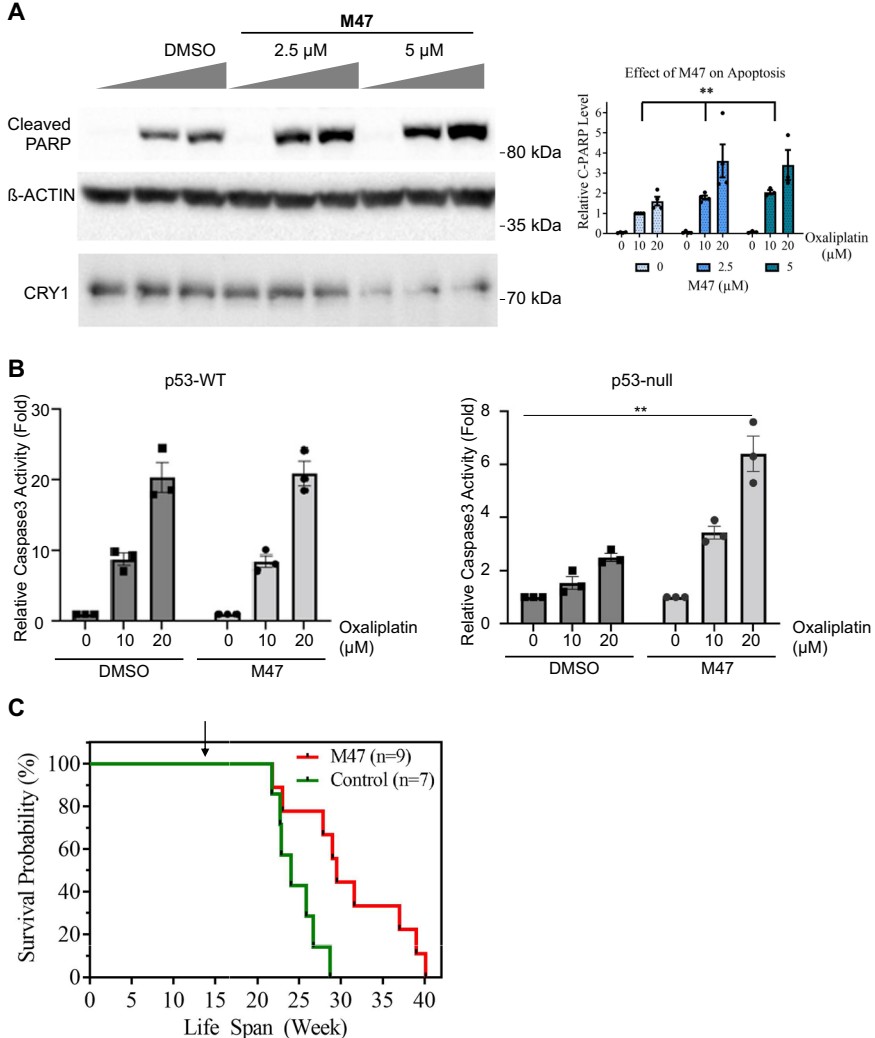

**Fig. 8 | The effect of M47 in *p53* null mouse skin fibroblast (MSF) cells in the presence of oxaliplatin and on the survival of *p53* mutant mice. A** Ras pT24 transformed *p53* null MSF fibroblast cells were treated with DMSO or M47 (5 μM) for 24 h and then 0, 10, 20 μM oxaliplatin for 16 h. Cells were lysed, and protein levels were analyzed by Western blot. In each case, M47 increases the cleaved PARP protein level and decreases CRY1 levels. Bar graph was drawn normalizing to 10 μM dosage of oxaliplatin (Data represent the mean ± SEM, $n = 4$, \*\*$p = 0.0051$ versus DMSO control by two-way ANOVA). **B** Measurement of Caspase-3 activity in WT or *p53*-null Ras-transformed MSF cells (*p53*-null) treated with oxaliplatin in the absence (DMSO) and presence of M47. Cells were treated with DMSO or M47 (5 μM)

for 24 h and then 0, 10, 20 μM oxaliplatin for 16 h. Caspase-3 activity in total cell lysates was measured using the Caspase-3 substrate, Z-DEVD-R110. The activity was normalized to total protein concentration and presented as fold activity to 0 μM treated samples. The results are the average of 3 biological replicates ± SEM (\*\*$p = 0.0077$ versus M47 treated cells with one-way ANOVA with Dunnet's post hoc test). **C** M47 in *p53*[-/-] mice increases age-adjusted survival. Kaplan–Meier survival analysis (log-rank test) of the time of death with evidence of tumors showed significant differences between vehicle-treated *p53*[-/-] and M47-treated *p53*[-/-] (\*\*$p = 0.0068$) ($n = 9$ for vehicle-treated and $n = 7$ for the M47 treated). Arrow indicates M47 administration (18th week).

Notably, knockout or knockdown of the *Cry*s in *p53*[-/-] mice or cell lines improved sensitivity to cancer chemotherapy by activating tumor suppressor genes[13,59]. In addition, overexpression of *PER2* genes in human pancreatic cancer cells prevented cell proliferation, initiated apoptosis, and behaved synergistically with cisplatin[60] thereby suppressing tumor growth. According to findings on the function of *Cry* and *Per2* in cancer, CRY destabilizers, which are predicted to be *Per2* enhancers, can be used as novel anticancer therapies. Since M47 decreases CRY1 and enhances the level of *Per2* both in the cell line and in vivo, we hypothesized that M47 could be a useful molecule for cancer treatment by promoting apoptosis. Thus, we analyzed the effect of M47 on the Ras-transformed *p53*-null fibroblast cells treated with oxaliplatin as a UV-mimetic. The enhanced effect of oxaliplatin upon M47 treatment implies that M47 can be utilized as an anticancer agent to increase the effect of oxaliplatin. Finally, we showed that administrating the M47 to *p53*[-/-] mice increased their life span by ~25%. This result is consistent with the previous study, where knocking out

*Cry* genes in the *p53* null mutant improved the lifespan of *Cry* mutant mice[13]. Collectively, our data suggest M47 has great potential to be used as a chemotherapeutic molecule for cancer types having *p53* mutations.

## Methods

### Molecular dynamics simulation

CRY1 (PDB ID:4K0R) and CRY2 (PDB ID:4I8G) structures were retrieved from the protein data bank. CRY structures were solvated with TIP3P water in a rectangular box by using NAMD (v. 2.13) and VMD (v. 1.9.1) programs as previously reported[61,62]. Minimization and heating protocols were followed on the solvated protein. The system was brought up to physiological temperature gradually by 10 K increase and simulated for 10 ps at each step. In molecular dynamics (MD) simulations, CHARMM-PARAM22 was used as a force field. Up on equilibration of the system, it was simulated for 300 ns at 310 K. Langevin piston was used to manage the pressure during the MD simulation. 2 fs time step

was used in the simulations. To calculate the total force on the system, the followings were done: particle-mesh Ewald (PME) method was used for long-range electrostatic interactions, van der Waals interactions were calculated with 12 Å cutoff, and bonded interactions were taken into account. For docking calculations, CRY1 structure (equilibrated) obtained from the MD simulation was selected. RMSD plugin of VMD was used to calculate the root-mean-square deviation (RMSD) of C, Cα, and N (backbone) atoms for each amino acid. Translational motions were excluded during RMSD analysis.

To further understand the selectivity of M47 for CRY1, the following methods were used: Homology models for CRY1 (3–491 aa) and CRY2 (21–510 aa) PHR domains were generated via RaptorX webserver to complete missing residues as described previously[25,63]. After equilibration, a production run of 300 ns for each system was performed using NAMD (v.2.13) software. The root-mean-square deviation (RMSD) calculations were performed using VMD RMSD utilities. Backbone atoms (C, Cα, and N) of each residue were used for RMSD calculation by excluding their translational motions. Conformational ensemble clustering and determination of cluster representatives were done using UCSF Chimera v.1.14. For dihedral angle calculations, all frames from trajectories were converted to pdb files with 1 ps intervals and dihedral angles were calculated with in-house python script.

## Molecular docking

For docking, more than 8 million small molecules with unknown functions were selected as ligands. These molecules are available through the web-site of Ambinter (Orleons, France). To eliminate irrelevant molecules, we filtered them according to the given properties: H-bond donor atoms should be less than 7, H-bond acceptor atoms should be less than 12, the molecular weight of molecule should be less than 600 Da, logP of molecule <7, the molecule should have at most 8 rotatable bonds, the molecule should have 3 or more aromatic rings, and totally should have 4 or more ring structures. To prepare ligands for docking, Openbabel 2.4.1, Autodock4.2, Autodock Tools4[64] and Autodock Vina v.1.2[65] programs, free for academics, were used. Finally, Autodock Vina program was used to dock nearly 2 million ligands to the target pocket on CRY1. According to the CRY-FBXL3 structure, the FAD and FBXL3 binding pocket was selected as the target pocket[66]. For visualization of protein structures and docking results, PyMol 2.5.0 (open source) (http://pymol.sourceforge.net/) and Autodock Tools4 were utilized, respectively.

Biotinylated M47(bM47) was drawn in Marvin Sketch. For molecular docking of M47 and biotinylated M47(bM47), aligned frames from CRY1 PHR domain simulation were extracted with 10 ps intervals. M47 and bM47 were docked to every extracted frame. Best possible binding poses were represented for each. The most crowded cluster representative in CRY2 PHR simulation was superimposed to the CRY1-M47 complex from molecular docking. Residues that were causing steric clashes were reported.

## MTT-toxicity assay

Cytotoxicity analyses were conducted on U2OS and NIH 3T3 cell lines (kindly provided by Prof. John Hogenesch Group). First, 4000 cells/well were seeded to clear 96-well plates (in triplicate). Let them grow for 48 h. Next, molecule treatment at determined doses was performed by not exceeding 0.5% DMSO as the final concentration. Molecules and cells were incubated for another 48 h. Since mitochondria convert tetrazolium dye 3-[4,5-dimethylthiazol-2-yl]−2,5 diphenyl tetrazolium bromide (MTT) to insoluble formazan (purple color), each well was treated with tetrazolium MTT to measure the cell viability. After incubating cells with MTT reagent for 4 h, the purple formazan was dissolved in DMSO:EtOH (1:1, v/v). With the help of the spectrophotometer, the absorbance of solutions in wells was measured at 600 nm. As a negative control, cells treated with 5% final DMSO were evaluated.

## Real-time bioluminescence monitoring

Two different machines were used for real-time bioluminescence assay: Synergy H1 (BioTek) suitable for 96-well plate readings was used for initial screening of molecules against circadian period; and Lumi-Cycle (Actimetrics, Inc., Evanston, IL) (for 35 mm plates) was used for high-resolution data. For Synergy H1, 50 000 U2OS cells stably expressing *Bmal1*-d*Luc* cells (kindly provided by Prof John Hogenesch Group) were seeded on an opaque 96-well plate. The next day, cells were treated with dexamethasone (DXM) (0.1 μM final) for 2 h for resetting. After 2 h, the media of cells were replaced with bioluminescence recording media having DMEM powder (sigma D-2902, 10X 1 L), 0.35gr sodium bi-carbonate (tissue culture grade, sigma S5761), 3.5gr D(+) glucose powder (tissue culture grade, sigma G7021), 10 mL 1 M HEPES buffer (Gibco 15140-122), 2.5 mL Pen/Strep (100ug/ml), 50 mL 5% FBS (in 1 L). Luciferin (0.1 mM final) was added fresh to the bioluminescence media. Molecules at determined concentrations were supplied to the recording media (0.5% as the final DMSO). To prevent evaporation and gas exchange, and hence to maintain cell homeostasis, optically clear film was used to seal plates.

In Synergy H1 luminometer, luminescence readings were recorded at 32 °C for each 30 min (with 15 s integration time) during one-week. In experiments using LumiCycle, $4 × 10^5$ cells (U2OS *Bmal1-dLuc* or NIH3T3 *mPer2-dLuc*) were seeded and the steps given above were followed as previously described[47]. For sealing the plates, vacuum grease was used and then placed into LumiCycle. Each plate was continually recorded every 10 min for 70 s at 37 °C by the help of the LumiCycle Software. Period and amplitude values were calculated by LumiCycle Analysis software v3.002 (Actimetrics, Inc., Evanston, IL).

## CRY-LUC cloning

The coding sequences of mouse *Cry1* and firefly Luciferase (Luc) (in pG5Luc plasmid, Addgene, Watertown, USA) were amplified by using primers with EcoRV-NotI flanking sites for *Cry*s and NotI-XhoI flanking sites for *Luc* (Table S1). Then, *Cry1-Luc*, *Cry2-Luc* and *Luc* genes were subcloned into the pcDNA4A-myc-his plasmid. After verifying the size of the product in agarose gel, NucleoSpin PCR and Gel purification kits (Macherey Nagel, Düren, Germany) were used to isolate the product from agarose gel. pcDNA4A plasmid with *Luc* (insert) and *Cry1* or *Cry2* pcDNA4A-myc-his (host) were digested by using NotI and XhoI enzymes (FastDigest, Thermo Scientific, Waltham, USA). To inhibit self-annealing, FASTAP enzyme (Thermo Scientific, Waltham, USA) was added to host plasmids. Inserts were isolated from the gel and then cleaned up by Gel purification kit (Macherey Nagel, Düren, Germany). Finally, T4 DNA ligase (Thermo Scientific, Waltham, USA) was used to ligate the insert and host.

## Site directed mutagenesis

For site-directed mutagenesis, quick-change method was utilized as reported previously[67]. We designed primers as 30 base pairs long in which changes in the base were placed in the middle. Sequences of all the primers used in site-directed mutagenesis are listed in Table S1. The PCR reaction mixtures have the followings: 0.3 mM dNTP, 10X Phusion GC Buffer (5 μl), (Thermo Scientific, Waltham, USA), DMSO (3%), primers (1 μM), template plasmid (m*Cry1* in pcDNA4A, 50 ng) and Phusion DNA polymerase (1 unit). Final volume is 50 μl. PCR was performed under the following conditions for 18 cycles: 98 °C for 30 S, 55 °C for 30 S and 68 °C for 5 min. We visualized amount of PCR products under UV after running them in 1% agarose gel. The samples having correct bands in the gel were digested with FastDigest DpnI enzyme (1U) (Thermo Scientific, Waltham, USA) at 37 °C for 1 h. 5 μl of DpnI-treated samples were transformed to *E.coli* DH5α cells. The next day, multiple colonies were picked and cultured. Then, we used miniprep (Macherey Nagel, Düren, Germany) to isolate plasmids. To confirm mutations in plasmids, Sanger sequencing (Macrogen, Netherlands) was performed.

## CRY-LUC degradation assay

For CRY degradation assays, we used *Cry1-Luc* and *Cry2-Luc* plasmids as reported[68]. In summary, *Cry1-Luc*, *Cry2-Luc*, mutant *Cry1-Luc* plasmids (40 ng) or *Luc* plasmid (5 ng) were sent to HEK293T cells ($4 \times 10^4$) (provided by Prof. Aziz Sancar) in an opaque 96-well plate (flat bottom) by reverse transfection with polyethylenimine (PEI). The next day, molecules or an equivalent amount of DMSO were added to cells. 24 h after the treatment, luciferin (0.4 mM final) and HEPES (10 mM final, pH = 7.2) were added to the cells. Cells were treated with cyclohex-imide (CHX) (20 μg/ml final) after the addition of luciferin, to inhibit protein synthesis. Optically clear film was used to seal the plate. Synergy H1 recorded the bioluminescence readings at 32 °C every 10 min for 24 h. One-phase exponential decay function of GraphPad Prism v.5.0 software (CA, USA) was used to calculate the half-life of proteins. In each experiment, at least three biological replicates were run for molecules and DMSO control.

## Protein immunoblots

We used RIPA buffer (50 mM Tris, 150 mM NaCl, 1% Triton-X, 0.1% SDS) to lyse the cells. At each time, a protease inhibitor cocktail (PIC) was freshly added to the buffer. Lysates were centrifuged at 4 °C, $12{,}500 \times g$ for 15 min to obtain the supernatant. Pierce Protein Assay (Thermo Scientific, Waltham, USA) was used to determine protein concentration as described in the manufacturer's manual. Dounce homogenizer with RIPA buffer was used to homogenize and lyze liver samples of mice. Upon homogenization, samples were chilled on ice for 10 min, and then centrifuged at $12{,}500 \times g$ for 15 min at 4 °C. The obtained supernatant after the centrifuge was evaluated as the whole cell lysate. To obtain subcellular fractions of liver samples, the followings were done: cytosolic lysis buffer (10 mM HEPES pH 7.9, 10 mM KCl, 0.1 mM EDTA, 0.05% NP40, proteasome inhibitor) was used to homogenize the samples. After pipetting the liver in cytosolic buffer 40 times with a cut tip, samples were chilled on ice for 10 min. Then samples were centrifuged at $700 \times g$ for 3 min at 4 °C. The supernatant was pipetted to a new tube and centrifuged at 4 °C, at $12{,}500 \times g$ for 3 min and the upper part of the solution was stored as the cytosolic fraction. The pellet was further treated to obtain the nuclear fraction. To get rid of any residues from the cytosolic part, the pellet was washed with cytosolic buffer. Samples, then, centrifuged at 4 °C for 3 min at $700 \times g$. The supernatant was removed, and the remaining pellet was lyzed in the nuclear lysis buffer (20 mM HEPES pH 7.9, 0.4 M NaCl, 1 mM EDTA, 10% glycerol, protease inhibitor) by pipetting. Then lysate was soni-cated with 60% power, 2 times, 10 s × 3 cycles. Samples were cen-trifuged at 4 °C, at $12{,}500 \times g$ for 5 min. The supernatant was stored as the nuclear fraction. The protein amount was determined as explained above. Then lysates were supplemented with 4X Laemmli buffer (277.8 mM Tris-HCl pH: 6.8, 4.4% LDS, 44.4% (w/v) glycerol, 0.02% Bromophenol blue, (freshly added) 5% volume of beta-mercaptoetha-nol) and incubated at 95 °C for 10 min. Samples were run in SDS-PAGE and then transferred to PVDF membrane (Millipore, USA) via wet transfer. 5% milk solution in 0.15% TBS-Tween 20 was used to block the membrane for 1 h. Membrane was incubated overnight at 4 °C or 1 h at room temperature with the following primary antibodies: CRY1 (Bethyl Catalog No. A302-614A; 1:5000 dilution), CRY2 (Bethyl Catalog No. A302−615A; 1:5000 dilution), CLOCK (Bethyl Catalog No. A302−618A; 1:2500 dilution), BMAL1 (Santa Cruz, sc-365645; 1:1000 dilution), anti-Myc (Abcam, ab18185; 1:5000 dilution), anti-His (Santa Cruz SC-8036; 1:10000 dilution) Beta-Actin, (Cell Signaling, 8H10D10; 1:10000 dilu-tion), Alpha-Tubulin (Sigma, T9026; 1:5000 dilution), HistoneH3 (Abcam, ab1791; 1:10000 dilution). After incubating with primary antibodies, membranes were washed with 0.15% TBS-Tween 3 times for 5 min. Membranes, then, were incubated with secondary antibodies conjugated with HRP. The followings were used as secondary anti-bodies: mouse (Santa Cruz SC-358920; 1:10000 dilution) or rabbit (Cell Signaling, 7074; 1:5000 dilution). To visualize the chemiluminescence

from HRP, we used ECL buffer system (Advansta WesternBright) to visualize HRP chemiluminescence (BioRad ChemiDoc Touch). Western blot images were quantified with the aid of Biorad Image Lab 6.1.

## RNA isolation and real time (RT) quantitative PCR (qPCR)

With the help of QIAGEN RNeasy Mini Kit (Hiden, Germany), we iso-lated the mRNA by following the manufacturer's protocol. Samples were treated with DNase supplied in the kit (on-column treatment). For liver samples, initially, samples were homogenized with Dounce homogenizer, and then the protocol given by the manufacturer was followed. To assess the quantity of isolated RNA, we used Nano-drop2000 (Thermo Scientific). We run isolated RNAs by running in ~0.5% agarose gel prepared by using DEPC water. Next, the gel was visualized under UV light. We converted ~400 ng of RNA from each sample to cDNA by performing first-strand cDNA synthesis by using MMLV-reverse transcriptase (Thermo Scientific, Waltham, USA). We performed this by following the given steps: 2 μl Oligo(dT)$_{23}$ and RNAs were mixed, then 1 μl 10 mM dNTP mix and nuclease-free $H_2O$ were added (final volume of 10 μl) (mix1). To denature any possible sec-ondary structures of RNA and primers that can prevent long cDNA products, this mix (mix1) was placed at 65 °C for 5 min. Then, mix1 was placed at 4 °C for 5 min. We next prepared 10 μl mixture2 (mix2) that has 10X reverse transcription buffer (2 μl), Ribolock RNase inhibitor (0.2 μl), Revert Aid Reverse Transcriptase (1 μl) and DNase/RNase free $H_2O$ (6.8 μl). Mix2 was mixed with mix1. The final mixture (20 μl) was placed at 42 °C for 1 h. To inactive the enzyme, the samples were incubated at 65 °C for 20 min. Upon completion of the cDNA synthesis, we added 80 μl of DNase/RNase-free $H_2O$ (final volume equals to 100 μl). To use cDNA samples in quantitative real-time PCR (qRT-PCR), we further diluted samples by 5-fold. SYBR Green was used to calculate mRNA expressions in qRT-PCR in which *GAPDH* was assessed as an internal control. A qRT-PCR reaction in final volume of 20 μl was pre-pared by mixing the followings: SYBR Green (8 μl), cDNA (3 μl, from 5-fold diluted sample), mix of forward and reverse primers (1 μl, 0.5 μM final), nuclease-free $H_2O$ (8 μl). All reactions were conducted in three biological replicates. In each replicate, two technical replicates were carried out. We used $2^{-\Delta\Delta CT}$ method to calculate the results of qRT-PCR that were reported relative to the *GAPDH* level in each sample. ΔCT values were accessed from Biorad CFX manager 3.1. We used the fol-lowing protocol for qRT-PCR reactions with 35 cycles of 95 °C 10 s for denaturation, 57 to 62 °C for 20 s for annealing (the degree is deter-mined by primer type) and 72 °C for 30 s for elongation. A list of pri-mers for qPCR was given in Table S1 indicated as RT_GeneName.

## Investigation the interaction between M47 and CRYs

In a 10 cm plate, $3 \times 10^6$ HEK 293 T cells were transfected with 10 μg of pcDNA4A-*Cry1*, pcDNA4-*Cry2* or pcDNA4A-*Cry1*-T5 plasmids. Each construct has His and MYC tags at their C-terminals. Cells were har-vested after 48 h of transfection. Cells were lysed in a lysis buffer having the followings: 50 mM Tris pH 7.4, 2 mM EDTA, 1 mM MgCl$_2$, 0.2% NP-40 (v/v), 0.1% sodium deoxycholate (w/v), 1 mM sodium orthovanadate, 1 mM sodium fluoride, and protease inhibitor cocktail (Thermo Scientific, USA). Lysates were pipetted and chilled on ice for 10 min. Vortexed briefly, and then chilled on ice 10 min. Then lysed samples were centrifuged at 4 °C for 10 min at $7000 \times g$. For input analysis, 5% of the supernatant was stored. The remaining lysates were separated into three and 2x binding buffer having 100 mM Tris-HCl pH 7.4, 300 mM NaCl, 0.2% NP-40 (v/v), 2 mM sodium orthovanadate, 2 mM sodium fluoride, and protease inhibitor was added to each solution with a 1:1 ratio. Each tube was supplemented with either of the followings: DMSO, biotinylated molecule, biotinylated M47 (synthe-sized by Enamine, Ukraine) and the competitor. The final mix was incubated with agitation for 2 h. During agitation, NeutrAvidin Agarose resin (Thermo Scientific, USA) was activated with the lysis buffer. Resins and lysis buffer were rotated for 5 min at 4 °C three times. End

of each rotation, resins were centrifuged at 4 °C for 2 min at 2500 × g. The resins were saved. After 2 h of mixing the lysates, the mixture was added to the activated resins. Resins and lysates were rotated for 2 h to isolate the proteins binding to the molecule. After 2 h of rotation, tubes were centrifuged (2500 × g, 2 min, 4 °C). We saved the beads and git rid of supernatants. Beads were washed with 1x binding buffer six times. Each time supernatant was discarded, and resins were retained. Finally, to isolate proteins from resins we added 50 μl of 4X Laemmli buffer and boiled them at 95 °C for 10 min. Western-blot technique was used to analyze pulled-down samples.

## Ubiquitination assay

1200 ng of pcDNA4A-*Cry1*-His-myc, 300 ng pBIND-*Fbxl3*, and 200 ng of p*hUb-Ha* were transfected to $5 \times 10^5$ HEK293T cells via PEI transfection reagent on 6-well plate. Cells were treated with DMSO or M47 after 28 h of transfection. After 42 h of post transfection cells were treated with MG132 to block proteasome dependent degradation or equal volume of DMSO to negative control cells. 48 h after transfection, cells were harvested. CRY1 proteins were purified using Myc resin (EZview™ Red Anti-c-Myc Affinity Gel, SIGMA-ALDRICH Catalog No: E6654) according to the manufacturer's instruction. In short, cells were lysed with RIPA buffer (50 mM Tris, 150 mM NaCl, 1% Triton-X, 0.1% SDS) and mixed with equilibrated myc-resins and agitated for 2 h at 4 °C. After washing three times with RIPA buffer, proteins were eluted with 4X Laemmli buffer (277.8 mM Tris HCl pH: 6.8, 4.4% LDS, 44.4% (w/v) glycerol, 0.02% bromophenol blue, freshly added 5% beta Mercaptoethanol). Ubiquitination and expression levels of CRY1 was detected by Western blotting with anti-HA (Santa Cruz Biotechnology, sc7392; 1:500 dilution) and anti-Myc (Abcam, ab18185; 1:5000 dilution) antibodies, respectively. Anti-Gal4(DBD) (Santa Cruz, sc-577; 1:1000 dilution) was used to detect Gal4-FBXL3 levels in the input.

## The effect of M47 on apoptosis

For apoptosis analyses, we measured Caspase-3 activity and cleaved PARP level under different concentration of the oxaliplatin. Caspase-3 activity was measured with an EnzChek Caspase-3 Assay Kit #2 (Thermo Fisher Scientific, USA) by following the manufacturer's instructions. Briefly, Ras pT24 transformed *p53*-null MSF fibroblast or wild-type MSF cells (kindly provided from Prof. Aziz Sancar Group) were treated with M47 (5 μM) or DMSO for 24 h, and then oxaliplatin was added at 0, 10, and 20 μM concentration for a 16 h incubation. At the end of this incubation, cells were pelleted by scraping and spinning down in microcentrifuge tubes. Pellets were lysed with 50 μL of cell lysis buffer and centrifuged at 1800 × g for 5 min. The supernatants were then mixed with Z-DEVD-R110 substrate and kept at room temperature for 30 min in the dark. Fluorescence intensities were measured at the excitation wavelength of 496 nm and the emission wavelength of 520 nm using Synergy H1 BioTek Multimode plate reader (BioTek Instruments, Inc., CA, USA). Control determinations were made on extracts of untreated cells. Each assay was performed in triplicate. BCA Protein Assay Kit (ThermoFisher Scientific Inc., USA) was used to determine protein concentrations and normalization. Cleaved PARP level was measured as described in Ozturk et al.[13]. Briefly, Ras transformed *p53*-null MSF cells were grown overnight and treated with (0-2.5-5 μM) M47 for 24 h. Cells were treated with oxaliplatin (0-10-20 μM) for 16 h. Then, cleaved PARP level was measured using anti-PARP (Cell Signaling 9542, USA; 1:1000 dilution). Beta-Actin levels as loading control were used for the normalization of cleaved PARP signals between the samples.

## Generation of CRYs knockout U2OS cell lines

CRY1 and CRY2 genes were targeted in U2OS cell line using the LentiCRISPRv2 system[68]. LentiCRISPRv2-CRY1 construct which was described previously[69]. Briefly, the annealed oligos (Sense: 5′ CACCG CCTTCAGGGCGGG GTTGTCG 3′; Antisense: 5′ AAACCGACAACCCCG

CCCTGAAGGC 3′) to target CRY1 and (CRY2 Sense: 5′CACCGCTGC-GACTCCACGACAA CC 3′; Antisense: 5′ AAACCGGTTGTCGTGGAGTC GCAGC 3′) to target CRY2 were inserted into BsmBI site of Lenti-CRISPRv2 plasmid (Addgene #: 52961).

The lentivirus preparation, transduction of U2OS cells and selection of the knockout candidates with puromycin (at 0.5 mg/mL concentration) were performed using a previously published protocol[69,70]. Cell lysates of the knockout candidates were analyzed for CRY1, CRY2 and Actin by immunoblotting using the following antibodies: anti-CRY1 (A302-614A, Bethyl Labs Inc. Montgomery, TX., USA, 1:1000 dilution), anti-CRY2 (A302-615A, Bethyl Labs, 1:1000 dilution), and anti-Actin (CST- 4967 S, Cell Signaling Technology, Boston, MA, USA, 1:1000 dilution). CRY1 and CRY2 immunoblotting showed the absence of the relevant CRY (CRY1 or CRY2) gene while the other CRY (CRY2 or CRY1, respectively) was present in that sample. The absence of both CRY1 and CRY2 was shown in CRY-DKO cell line. Actin immunoblotting was used as a loading control. HRP-labeled anti-mouse and anti-rabbit antibodies (Cell Signaling Technology) were used at 1:5000 dilution. Chemiluminescence was developed using WesternBright Sirius HRP substrate (Advansta, San Jose, CA, USA, cat no: K-12043-D20). Images of immunoblots were captured using the ChemiDoc XRS + system (Bio-Rad).

To detect the effect of M47 in the absence of CRYs, $3 \times 10^5$ *Cry1*, *Cry2*, and *Cry1/Cry2* knockout (CRY-DKO) U2OS *Bma11*-d*Luc* cells were seeded in 35 mm clear plates. Then, medium was changed with Lumi-Cycle medium having DMSO or M47, sealed with vacuum grease, and placed to LumiCycle.

## Investigation of the off-target effect of M47 with mass spectrometry analysis

The whole cell lysate was prepared with ice-cold lysis buffer (50 mM Tris-HCl, pH 7.4, 2 mM EDTA, 1 mM MgCl₂, 0.2% Nonidet P-40, 1 mM Na₃VO₄, 1 mM NaF and protease inhibitor mixture) from HEK 293 T. Lysate was diluted with ice-cold 2 × binding buffer (100 mM Tris-HCl, pH 7.4, 300 mM NaCl, 0.2% Nonidet P-40, 2 mM Na₃VO₄, 2 mM NaF, and protease inhibitor mixture) and divided into three fractions to be treated with either DMSO, or 100 μM bitoinylated-M47 (bait) with and without 500 μM M47(competitor). After equilibrating NeutroAvidin-agarose resin (Thermo Scientific, catalogue number 29,200) with lysis buffer, the lysate-compound mixture and NeutroAvidin-agarose resin were incubated and mixed continuously at 4 °C for 2 h. The beads were washed four times with 1 × binding buffer. To elute the bound proteins Laemmli buffer was added, and samples were heated at 95 °C for 7 min. The pulldown precipitates were subjected to SDS-PAGE and transferred to a polyvinylidene difluoride membrane (Millipore). Anti-MYC was used for MYC-tagged CRY1 and CRY2 proteins. The LC-MS/MS examination of the pulldown precipitates (Thermo Proteome Discoverer versions 1.4 and 2.3) was performed at Koç University's Proteomics Facility (Istanbul, Turkey) as previously described[15,71].

## In vivo studies

**Animals and their synchronization.** The in vivo studies with M47 experiments are carried out using C57BL/6 J male and female mice that were 8–12 weeks old and weighed 18–24 g. Mice were purchased from the Koc University Animal Research Facility (Istanbul-Turkiye). This study was approved by the Koç University Animal Research Local Ethics Committee (No: 2015/13). A room, controlled humidity (55 ± 5%) and temperature (21 ± 2 °C) was utilized to house up to four mice in polystyrene cages. All the animals were housed in the specific pathogen-free at the Animal Facility Koç University. The animals were kept in a room where they exposed to exposed to a 12-h light/12-h dark cycle and provided food and water *ad libitum*. These ambient conditions are described in Zeitgeber time (ZT), where ZT0 corresponds to lights on and ZT12 to lights off. Mice received either M47 or vehicle i.p. at ZT3. Animals were euthanized by cervical dislocation or carbon dioxide.

**Humane endpoints.** For animal welfare, animals were monitored on a daily basis and examined for following indications: body temperature, body weight changes, clinical signs, water and food consumption, and behavior. For ethical concerns, animals were sacrificed after observing excessive reduced locomotor activity (inability to access food and water), unbearable tumor load (>10% of body weight or >1.5 cm diameter), excess weight loss (>20% of body weight), dyspnea, and excessive hypothermia (<33 °C).

**M47 formulation.** Before being injected i.p., M47 was freshly prepared by being dissolved in 2.5% DMSO and 15% Cremophor EL before being diluted with 82.5% isotonic sodium chloride solution (Vehicle = DMSO:Cremophor EL:0.9% NaCl; 2.5:15:82.5, v/v/v). Unless otherwise noted, all solvents were reagent grade, and all other commercially available reagents were used as received.

**Dose range determination.** The OECD Guidelines were followed in choosing the dose levels to be employed in the single dose toxicity research (Guidelines for the testing of chemicals, 2002). At each dose, C57BL/6 J male and female mice (8–12 weeks old) were used ($n = 2$–3). One dosage of M47 was administered i.p. to mice in each group at 5, 50, 300, or 1000 mg/kg. Control mice ($n = 3$ for each sexes) received vehicle treatment. For 14 days, mice were monitored for changes in body weight, body temperature, abnormal behavior and clinical conditions, and mortality. At the end of the study, all mice underwent a gross necropsy. Body weight was recorded as a measure of overall toxicity. The M47-induced change in body weight was expressed in relation to the body weight on the first day of treatment. For five days after receiving an injection of M47, the mice's body temperatures were monitored using a rectal homeothermic monitor (Harvard Apparatus, USA) at the same time in each day. For a period of 14 days, water and food intake were also measured. Mice were given isoflurane anesthesia following the completion of the studies, and blood was drawn through heart puncture. Following blood collection, mice were instantly scarified by cervical dislocation. Blood samples were examined for hematological markers.

**Administration in the repeated dose toxicity study-1: determining the maximum tolerated dose of M47 after a 5-day.** The findings from the single dose toxicity trial were used to help choose the doses for this repeated dose toxicity study. M47 was administered i.p. at 40, 80, or 150 mg/kg for five days to C57BL/6 J mice ($n = 6$ per group), aged 8 to 12 weeks. Vehicle (DMSO:Cremophor EL:0.9% NaCl; 2.5:15:82.5, v/v, i.p.) was the only treatment given to control mice ($n = 4$). Animals were observed for mortality, clinical symptoms, weight changes, body temperatures, dietary intake, behavior, and gross findings at the final necropsy throughout the course of the study. As a toxicity indicator, mice's body weights and temperatures were recorded daily. Mice were punctured in the heart while being sedated with isoflurane within 48 h of the last dose of M47 being administered. Blood was examined using hematological and biochemical methods. For histological analyses, the liver, spleen, kidneys, and lungs were taken and preserved in 10% formalin solution.

**Administration in the repeated dose toxicity study-1: determining the subacute toxicity of 60 mg/kg M47 after a 14-Day.** The dose of M47 for the following 14-day subacute toxicity research was established as 60 mg/kg based on the findings of the 5-day maximum tolerated dose (MTD) determination study. 10 C57BL/6 J mice ($n = 10$) between the ages of 8 and 12 weeks were administered an i.p. dosage of M47 at 60 mg/kg for 14 days. Vehicle was administered to control mice ($n = 5$). After that, the 5-day MTD determination research indicated in a previous section was carried out in the same manner.

**Pharmacokinetic studies.** A single 100 mg/kg i.p. dosage of M47 was administered to 8–12-week-old female C57BL/6 J mice ($n = 4$ at each time point). After administering M47, blood samples were taken 0, 0.5, 1, 2, 4, 8, 12 and 24 h later by heart puncture while under isoflurane anesthesia. Heparinized tubes were used to extract plasma, which was then centrifuged and kept at −80 °C until analysis. Brains from the animals were removed and stored at −80 °C for further process. M47 levels in brain were assessed at 2 h ($n = 3$) and 4 h ($n = 3$) in accordance with the highest plasma concentrations of M47.

**Assessment of M47 concentrations in plasma and the brain.** To determine M47 and the internal standard (IS) of loratadine, liquid chromatography coupled to tandem mass spectrometry (LC-MS/MS) was used to detect the analytes in plasma samples. Ion source parameters were set to be optimal for the analytes while the instrument was run in full scan mode. The electrospray ionization (ESI) source was operated under the following conditions: 300 °C for the gas, 10 L/min for the flow, 25 psi for the nebulizer, and 5500 V for the capillary voltage. For M47, ESI in the positive ion mode yielded the best results. Protonated molecules were chosen as precursor ions for MS-MS detection, and the most numerous fragment ions formed at collision energies of 22–20 eV were observed as the product ions for M47 and IS. Using mass transitions for M47 $m/z$ 472.1 → 353.4 [collision energy (CE) = 22 eV] and fragmentor 234 V, for IS $m/z$ 383.2 → 337.2 [CE = 20 eV] and fragmentor 140 V, and for IS, M47, IS, and M47, respectively, MS-MS analysis was carried out in the selected reaction monitoring (SRM) positive ionization mode. The analytes were separated using a standard reversed-phase liquid chromatographic separation on $C_{18}$-bonded silica with a mobile phase of methanol:% 0.1 formic acid (90:10, v/v). Methanol was used in liquid-liquid extraction to separate M47 from plasma samples. The LC-MS/MS system received 10 μL of material.

**Determination pharmacokinetic properties of M47.** The M47 plasma concentration-time curve was used to directly calculate the peak plasma concentration ($C_{max}$) and the time needed to reach it ($t_{max}$). The trapezoidal approach was used to determine the area under the plasma concentration-time curve from 0 to 48 h ($AUC_{0-48h}$). $AUC_{0-\infty}$ was calculated by adding the extrapolated component after the last sample time using standard procedures. The non-compartmental approach was used to determine the other pharmacokinetic parameters of M47. The slope of the line that was drawn after calculating the elimination rate constant ($k_{el}$) from the terminal points of the M47 plasma concentration-time plot was equal to $k_{el}$. Using the data's terminal points as an estimate, the terminal elimination half-life ($t_{1/2}$) was determined. The formula $t_{1/2} = \ln2 / k_{el}$ was used to convert between $t_{1/2}$ and $k_{el}$.

**Determination the survival curve in $p53^{-/-}$ mice in the presence of M47.** We obtained C57BL/6 J $p53^{+/-}$ mice from Jackson Laboratory (USA). To propagate $p53^{-/-}$ mice, $p53^{+/-}$ mice were bred. Mouse tail were used for PCR amplification with primers listed in Table S1. For 7 consecutive weeks, mice ($n = 9$) received 50 mg/kg/day of M47 i.p. on 5 days each week. Vehicle treatment was given to the control animals ($n = 7$). The repeated dosage toxicity tests' findings were used to determine the M47 dose. Daily mortality was recorded, and Kaplan-Meier survival analysis with log-rank test was performed using GraphPad Prism version v.8.0 (GraphPad Software, CA, USA).

**Statistical evaluation.** For each analyzed variable, the data were reported as means ± standard error of the means (SEM). Different versions of GraphPad Prism for Windows were used for the statistical analyses (GraphPad Software, CA, USA). The Student's t-test and one- or two-way analysis of variance (ANOVA), performed after the Tukey or Bonferroni post hoc tests, respectively, were used to confirm the

statistical significance of differences between groups. The p-values for each significant test employed in each experiment were listed in the figure legends.

## Reporting summary
Further information on research design is available in the Nature Portfolio Reporting Summary linked to this article.

## Data availability
All other data generated or analysed during this study are included in this published article (and its Supplementary Information files). Source data are provided with this paper. Mouse CRY1 (PDB ID: 4K0R) and mouse CRY2 (PDB ID: 4I6G) are used in this study. The proteomics data generated in this study have been deposited in the PRIDE database under accession code PXD037311. Source data are provided with this paper.

## Code availability
Phyton scripts included in this published article in the source data.

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

## Acknowledgements

This work was supported by a TUBITAK SBAG (215S021) grant and an Istanbul Development Agency grant (ISTKA-TR/14/EVK/0039).

## Author contributions

S.G. designed the study and performed the experiments and wrote the manuscript. Y.K.A., Na.O., and A.O. perform all mice related experiments. Y.K.A. and A.O. wrote the manuscript. T.K., F.A. and Nuri.O. performed apoptosis and gene-editing. S.G., S.I., S.S. and Z.M.G. performed in vitro studies. I.D. and D.O.U. performed the analytical quantifications from plasma/tissue. Nurhan.O. and B.A. performed proteomics studies. O.O. performed computational studies. A.C.T. handled all mice. O.S.I. and M.G. analyzed molecule, M.T. helped computational works. I.H.K. conceived experimental design and wrote the manuscript.

## Competing interests

The M47 studied in this paper is awarded of the following patent: WO2021137771A1 (World Intellectual Property Organization, Patent Cooperation Treaty). S.G., M.T., and I.H.K. are inventors in these patent applications, while Koc University is the assignee. The title of the patent is "11-(4-chlorophenyl)–4-(2,3-dihydro-1h-indole-1-carbonyl)–3,11-dimethyl-5,10,dioxatricyclo[7.4.0.0,2,6,]trideca-1,3,6,8-tetraen-13-one and derivatives as destabilizer of cry1 for the treatment of circadian rhythm associated diseases and disorders". The invention provided an anticancer molecule against cancers harboring p53 mutations. The other authors declare that they have no competing interests.
