## [Peer Review File · Nature Communications]

REVIEWER COMMENTS

Reviewer #1 (Remarks to the Author):

This study by Gul et al identified a small molecule M47 by virtual screening and molecular docking based on the mouse CRY1 protein structure. M47 was further characterized using various biochemical and molecular methods and is proposed to selectively destabilize CRY1 protein and have effects on circadian rhythms. The results of this study validated the previous findings that reducing CRY1 enhances apoptosis and increases the survival rate in a p53^{-/-} background. This study shows that structure-based virtual screening is a feasible and efficient approach for identifying novel CRY1 ligands. M47 is identified as an isoform-selective destabilizer, binding CRY1 only, not CRY2, facilitating the precise manipulation of CRY in mammals. Despite some of the potential in this study, there are still many major concerns and minor issues that need to be comprehensively clarified and addressed.

Major concerns:

1. It is difficult to interpret the Cry1/Cry2 KO data (Fig. 1C) without wild type MEF data. What is the effect of M47 on the Per2 reporter? According to the authors, CRY1 degradation by M47 is expected to release Per2 from repression (as shown in Figs. 4B and 6D), but in NIH-3T3 cells, Per2 reporter is repressed by M47 (Fig. S2D). The authors need to explain this discrepancy, in addition to providing wild type MEF data.

2. It is interesting that M47 enhances CRY1 ubiquitination and degradation. However, the mechanism is not clear. Does M47 mimic C-terminal tail of FBXL3 that is required for CRY degradation? The authors need to discuss about potential mechanism, in comparison with CRY-stabilizing compounds that were shown by X-ray crystallography to interact with the FAD binding pocket where M47 interacts.

3. While Cry1 KO results in period shortening, CRY1 degradation by M47 causes period lengthening. The authors suggested that M47 interaction with R293 may stabilize CRY1-CLOCK complex for period lengthening (Discussion). However, the stabilization of CRY1-CLOCK complex will enhance repression, resulting in reduction of target gene expression including Per2, but this was not the case (Per2 was increased by M47 in Figs. 4B and 6D). The authors need to explain this discrepancy.

4. Based on the docking pose (Fig. 2D), the position where the authors attached biotin linker (Fig. S3B) seems to be facing inside the pocket, making the binding pose questionable. The authors need to explain this point, at least by providing the best docking model of biotinylated M47 to CRY1, and by comparing it with M47.

5. Regarding the orientation of W292/W310 and W399/W417, the provided model (Figs. 3D and 3E) is incompatible with newer crystal structures of CRY1 and CRY2 apo forms (PDB IDs: 5T5X, 6KX4, 7D0M, and 7D0N) (for example, "CRY1" 5T5X and 6KX4 structures are similar to the "CRY2" model that the authors provided), as well as many compound-bound structures including CRY2-KL001 (PDB ID: 4MLP) and CRY1-KL001 (PDB ID: 7DLI) (i.e., the proposed orientation of the tryptophan residues interferes compound interaction in both CRY1 and CRY2). In addition, 50 ns simulation is too short to make conclusion, compared to the standard of the field, and can be largely affected by the initial structure. Together with the point mentioned in comment 4, the provided M47 binding mode and the mechanism of CRY1 selectivity are questionable. Mutation of the key pocket residues R293 and W399 could simply disrupt the FAD pocket to prevent M47 interaction (Fig. 2E). A recent extensive study of the orientation of these tryptophan residues and its role in compound selectivity (Miller et al., PNAS 118: e2026191118, 2021) would be informative.

6. To enable comparison with cellular assay data, please provide compound concentration in micro molar, but not ng/ml, in Figs. 5G and 5H, and Table 1. What is the expected concentration of M47 in the liver at 50 mg/kg injection in Fig. 6? Is it comparable with cellular assay data in Fig. 4 (by the way, M47 concentration information is missing in Fig. 4). Also, it is important to show CRY2 blot in Fig. 6 (and also Fig. 7A) as a control which is not affected by M47.

7. The specificity of M47 on CRY1 was mainly focused on the comparisons between CRY1 and CRY2. There is not enough evidence to exclude other off-targets beyond CRY2. Given Fig.2C as an example, M47 does not change the period length in CRY1 deficient cells, but the intensity (or maybe the amplitude) seems to be reduced. It would be worthwhile to consider also conducting an experiment to determine that period length and amplitude of luciferase reporter is similar to what is seen in Fig.1B in CRY2-/- Bmal1-dLuc cells. Considering M47 is not affecting the cell viability and LUC stability as shown in this study, what are the other targets causing the reduced intensity in CRY1 deficient cells? In addition, it is insufficient to use Cry1-/- Cry2-/- MEF Per2-dLuc bioluminescence (Fig.1C) to demonstrate M47'S specificity on CRY. Experiments such as biotinylated-M47 pull-down mass spectrometry analysis would be necessary to evaluate the off-targets in a broader scope.

8. One selection criteria for candidate molecules in this study was the cell viability in U2OS cells. It may not be appropriate to assess the toxicity in tumor cells such as osteosarcoma. Molecules toxic to tumors but nontoxic to normal cells are good for cancer therapy. The cell viability criteria could be done in normal healthy cells.

9. To demonstrate that the CRY1 specific effects of M47 on apoptosis and survival were not due to any other off-targets, it is better to include a CRY1-/- p53-/- control in studies of Fig.7.

10. CRY1 expression oscillates and the half-life of M47 is only some hours. In toxicity in vivo studies, the M47 injection time was 3hrs after light onset. In the survival study, M47 was injected 17hrs after light onset. Are the effects and toxicity of M47 time-dependent? Please justify the choice of different time points for M47 injection with necessary supporting evidence.

11. Animal experiments were sometimes done in both male and female mice, sometimes in females only, and sometimes the sex was not mentioned. Please clarify the sex of animal used in each experiment and specify whether there are sex effects if both sexes were used. Additionally, only 2 mice were used per hour of treatment in the brain exposure experiment. A minimum of 3 animals should be used.

12. The authors concluded in the end that "Our results suggest that reducing the level of CRY1 protects mice from tumor burden and increases the survival rate of p53-/- mice". Although the survival rate was shown, no experimental evidence regarding the protection from tumor burden was demonstrated in the results. Does M47 decrease the tumor or any histological changes after the injection of M47? Specific evidence supporting the conclusion should be presented.

13. The cleaved PARP WB method was used for evaluating the apoptosis. But WB is not a reliable quantitative assay. It would be better to include additional more reliable quantitative assays for apoptosis.

14. Authors state that the effect of M47 and selective targeting of CRY1 via M47 is promising dependent on p53 mutation status. No experiments were done to compare effects of treatment of M47 in p53+/+ cells or mice versus p53-/- cells or mice. Additionally, it would be worthwhile to see how treatment of M47 changes tumor suppressor gene expression, BMAL1::CLOCK downstream gene expression, and cell proliferation in both cancer cells and tumor tissue in mice.

15. Fig.3A: The input portion of the blot is separate from the other conditions. This should be one single blot.

Other minor points:

1. Is "data not shown" allowed?

2. Lines 59-60: Full names of BMAL1 and CLOCK were not written out in the introduction prior to the acronyms.

3. Line 109: It would be helpful to those not familiar with the structure of CRY1 to indicated that the FAD binding pocket is located in the PHR domain, which is why the screen was designed to find molecules that target the PHR domain. The introduction only mentioned the FAD-binding pocket. Can move the sentence in Lines 119-121 up to help with this.

4. Line 136: Why destabilized Luciferase was used, instead of standard Luciferase? Also, it is not clear whether the authors used dLUC or LUC in Figs. 2A and 2B. Should all LUC be dLuc (i.e. Figs S2)? If not, why were different luciferases used?

5. Some of the colors of the curves are hard to distinguish between different samples, for example, Fig.2A, B, and E are all blue and green lines, very similar. Better to use another color scheme. Also, in Fig.1C, what's the red line? The line patterns in the plot are not consistent with

the legend.

6. Fig. S2D: Period change in NIH3T3 Per2-dLuc cells is not clear. Quantified period data need to be provided. In addition, NIH3T3 Bmal1-dLuc was mentioned in the main text (line 145), but the data was not provided.

7. Authors do not explain what is attributing to the lower body weight in animals given 50 mg/kg M47 at Day 2 and 3 compared to animals given vehicle or 5 mg/kg M47 (Fig.3B).

8. Fig.4A, maybe the blot is not a representative one, but the CLOCK protein blots intensity does look higher with M47 than with DMSO.

9. Line 254: "next hat"?

10. Fig. 5A shows that body temperature is higher at 5 mg/kg, while the main text says 50 mg/kg but not 5 mg/kg is higher (lines 255-257).

11. Line 260: Should be changed to SDT

12. Line 283: Fig 4G should be Fig 5G?

13. Line 286: Remove h from "haccording".

14. Line 293: Remove th from "6th".

15. Fig. 6A legend, if not statistically significant, the null hypothesis (no difference) is accepted, and thus it is inappropriate to say "M47 slightly decreased the CRY1 levels in whole cell lysate (WCL) and cytosolic fraction".

16. Figs. 6B (cytosolic) and 6C (nuclear) are incorrectly cited in the main text (lines 298-301).

17. Do authors have an explanation as to why there isn't a dose dependent decrease in Per2 mRNA levels in Fig. 6D when there is a dose dependent decrease in CRY1 levels in Fig. 6C?

18. It is inappropriate to use t-test for the experiment with more than 2 groups (Figs. 1, 2, 4, 6).

19. Please indicate when the injection was started in Fig.7B.

20. The degree sign is not correct. Many of them are number 0, not a degree sign. When Kelvin (K) is indicated, a degree sign should not be included.

21. Lines 323-326: The two sentences are contradicting each other and are suggesting that C57BL/6J mice develop cancer at 20 weeks of age regardless of p53 status. Please make sure the information of the age and cancer incidence is correct.

22. Line 329: error symbols between NF and B.

23. Line 358: Full name of ROR was not written out prior to the acronym.

24. Line 433: "04" needs to be removed along with the extra spaces.

25. Line 468: The "s" in sec does not need to be capitalized.

26. Line 528: centrifugation 01 13000rpm?

27. Line 582: please indicate the cell number and plate size.

28. Line 602: Indicate number of cells used.

29. Line 609: remove + sign.

30. Please check discrepancies between h and hr (i.e. Line 608: "hr").

31. Lines 286 and 701: "2nd" and "4th" should be changed to 2 h and 4 hr.

32. Lack of a space after h in many "hafter" and "hof".

33. Some "in vitro" and "in vivo" are not italicized, while others are. Please keep consistent.

34. Some of the letter "l" in "Luc" is not capitalized, while most are. Please fix.

35. Ref. #14 and #35 are the same. Ref. #17 and #40 are the same. Ref. #31 and #52 are the same. Ref. #47 and #56 are the same.

Reviewer #2 (Remarks to the Author):

This is an interesting paper in which the authors report the identification of a molecule from a library of approximately 2 million small molecules. This compound, M47, was among the 200 molecules in the library that exhibited computationally high affinity to the FAD binding pocket of mouse CRY1. The authors first demonstrated that M47 is not too toxic to U2OS cells at physiologically relevant concentration. Then, they went on to demonstrate that M47 lengthened the period of circadian rhythm as determined by luciferase assay in derivatives U2OS and NIH3T3 cells. The drug exerts its activity by physically binding to CRY1, but not to CRY2. The authors present reasonable computational/structural evidence for this selectivity despite the fact that the

"FAD binding pocket" of the two CRYs have nearly identical sequences. Finally, after determining the toxicity and pharmacokinetic profile of the drug in mice the authors explore the chemotherapeutic potential of the compound by testing it in p53^{-/-} MEF and in p53^{-/-} mice. They find that it promotes apoptotic death of the p53^{-/-} Ras transformed MSF, and increase the lifespan of p53^{-/-} mice by 25%. The authors suggest that M47 is a potential candidate as an adjuvant in cancer chemotherapy. Overall the data that combine all state of the art approaches to identify novel anticancer drugs is convincing and the paper is a valuable contribution to the field. I have the following comments that need to be addressed in the revision:

1) In the introduction the authors note that M47 destabilizes CRY1 and increase the period length. This is contrary to Cry1^{-/-} mice. The authors, much later, in the Discussion section remark that in cell lines other small molecules that stabilize CRY1 in fact shortened period length (Ref 25) and thus, in cellulo effects do not necessarily replicate the Cry1 mutation effect in whole animal. This explanation should be given early on so the reader will not wonder to the very end about this apparent paradox.

2) "FAD binding pocket": This is defined by the crystal structure of photolyase [Park HW (1995) Science 268:1866]. In mammalian CRYs this pocket is wider and open to the solvent and for this and other structural reasons the mammalian CRYs have no FAD. Hence, either this point should be explicitly stated or the phrase should be in quotation marks "FAD binding pocket".

3) Line 137: "data not shown" should be shown in supplementary Figures (Tables).

4) Line 234: "slight change in BMAL1 level": It does not look small in Fig 4A. Could it be downregulation CRY1 leads to upregulation of NR1D1 which leads to downregulation of BMAL1.

5) Lines 303-304: "In agreement...decreased CRY1...[lead] to increased expression level of Per2..." is in accord with the finding of Vitaterna et al (1999) PNAS 96: 12115 which reports this effect first. This confirmatory data strengthens the authors' agreement. I am surprised the authors are not marking this comparison.

6) Lines 312-317 and Fig 7A: there is a confusion between MSF and MEF. To my knowledge Ras24 transformed p53 null cells are MSF. This should be clarified with an appropriate reference. I could not find a mention of this in the M&M section.

7) Minor point: At the highest concentration M47 seems to have a drastic effect on AMPLITUDE as well. A comment is needed.

8)The authors, in the introduction and at various sections of the paper mention clock disruption as carcinogenic and refer to some general reviews on the clock. In fact two exhaustive reviews published this year critically discuss the lack of convincing evidence for this claim. As a matter of fact this manuscript is further evidence that clock disruption is anticarcinogenic. So why make a blank statement for the claim to the contrary?

9) The writing contains numerous unusual sentences constructs commonly seen with non-native English-speaking authors. Perhaps this can be corrected by the editorial staff.

Reviewer #3 (Remarks to the Author):

The Manuscript by Seref Gul et al is well written and structured with clear and logical progression throughout the MS. Here, the author describe, in detail, the discovery of a cryptochrome 1 targeting capacity and thoroughly assess its capacity to destabilize this key regulatory molecule. A point by point assessment of this process is seen from in vitro to In vivo approaches and in particular assess it capacity in a cancer- like setting with emphasis on P53^{-/-} setting. While the authors assess this initially in fibroblast...which do no represent cancer as such they nicely show in a p53^{-/-} whole body setting that this expands life and since p53 is commonly lost in cancer this claim is valid and well carried out. I feel the MS is ready for publication with a few key aspects addressed.

Can the authors discuss the use of cell line in the study as many have p53 mutation rather than loss or intact Wt p53 and this is a critical caveat of the study...as GOF p53 is a key driver of cancer metastasis the author should at least discuss this and add these details to their study – for example USOS cells, NIH 3T3 cells, HEK 293 cells (WT for p53) thus activation of apoptosis is intact...can the authors perform experiments in a p53^{-/-} cancer cell lines in similar fashion to there initial experiments in fibroblast..... they can do this in a wt versus null setting and it is important to

know if this works in GOF mutant P53 settings.

RESPONSE TO REVIEWER COMMENTS

Reviewer #1 (Remarks to the Author):

This study by Gul et al identified a small molecule M47 by virtual screening and molecular docking based on the mouse CRY1 protein structure. M47 was further characterized using various biochemical and molecular methods and is proposed to selectively destabilize CRY1 protein and have effects on circadian rhythms. The results of this study validated the previous findings that reducing CRY1 enhances apoptosis and increases the survival rate in a p53^{-/-} background. This study shows that structure-based virtual screening is a feasible and efficient approach for identifying novel CRY1 ligands. M47 is identified as an isoform-selective destabilizer, binding CRY1 only, not CRY2, facilitating the precise manipulation of CRY in mammals. Despite some of the potential in this study, there are still many major concerns and minor issues that need to be comprehensively clarified and addressed.

Major concerns:

1. It is difficult to interpret the Cry1/Cry2 KO data (Fig. 1C) without wild type MEF data. What is the effect of M47 on the Per2 reporter? According to the authors, CRY1 degradation by M47 is expected to release Per2 from repression (as shown in Figs. 4B and 6D), but in NIH-3T3 cells, Per2 reporter is repressed by M47 (Fig. S2D). The authors need to explain this discrepancy, in addition to providing wild type MEF data.

We used *CRY1/CRY2* double knockout (DKO) U2OS *Bmal1-dLuc* cell line to assess the effect of M47 on reporter gene and to be compatible with other data, which we used U2OS cell line. Therefore, we replaced figure 2C with data obtained from *CRY1/CRY2* (DKO) U2OS. The new data clearly show that M47 had no dose dependent effect on the reporter enzyme under the control of *Bmal1* promoter. In addition, we made typo for NIH 3T3 *Per2-dLuc*. The actual cell line we used in this study is NIH 3T3 *Bmal1-dLuc*. This clarification was made in the text. The explanation of why M47 caused repressed reporter activity not only in NIH but also U2OS cells is provided in the text as follow:

“We also observed that M47 caused not only period shortening in these cell lines, but also a dose-dependent decrease in amplitude of rhythms. These results are consistent with previous studies where either reduction or deletion of the *Cry1* genes result in short-period and low-amplitude rhythms or even arrhythmicity in different cells including U2OS and mutant mice²⁹⁻³⁴. This may be due to fact that *Cry1* is essential for generation of cell autonomous circadian clock function³⁵. To eliminate the possibility that M47 might affect the other proteins and, in turn, circadian rhythmicity we tested the effect of M47 in *CRY1/CRY2* double knockout (DKO) U2OS *Bmal1-dLuc* (Fig. S1E)”

2. It is interesting that M47 enhances CRY1 ubiquitination and degradation. However, the mechanism is not clear. Does M47 mimic C-terminal tail of FBXL3 that is required for CRY degradation? The authors need to discuss about potential mechanism, in comparison with CRY-stabilizing compounds that were shown by X-ray crystallography to interact with the FAD binding pocket where M47 interacts.

We provided following explanation to highlight the potential mechanism between CRY1 and FBXL3 in the manuscript as follow:

“The last amino acid residue of the C-terminal tail of FBXL3, Trp428, is shown to be critical for the interactions with CRY and their ubiquitination²³. It is possible that indole group of this residue might interact with indole group of M47 by pi-pi interaction at “FAD-binding pocket” and in turn stabilize the interaction between CRY1 and FBXL3”

3. While Cry1 KO results in period shortening, CRY1 degradation by M47 causes period lengthening. The authors suggested that M47 interaction with R293 may stabilize CRY1-CLOCK complex for period lengthening (Discussion). However, the stabilization of CRY1-CLOCK complex will enhance repression, resulting in reduction of target gene expression including Per2, but this was not the case (Per2 was increased by M47 in Figs. 4B and 6D). The authors need to explain this discrepancy.

As we also discussed in the text that the period regulation by CRYs at cellular is a complex and is not well understood. Since we have no data regarding CRY1-CLOCK interaction in the presence of M47, we revised the section by deleting our CRY1-CLOCK interaction discussion.

“Despite a decrease in the level of CRY1, M47 lengthens the period and dampens the amplitude in U2OS *Bmal1-dLuc* cells. This is possibly due to the potent repressor activity of the CRY1 as shown previously³⁵. Cell-based genetic complementation studies also show that the stability and amount of CRY, *per se*, do not dictate the period length, but that the subcellular localization and interaction of CRY with other core clock proteins determine the period^{5,15,53-55}.”

4. Based on the docking pose (Fig. 2D), the position where the authors attached biotin linker (Fig. S3B) seems to be facing inside the pocket, making the binding pose questionable. The authors need to explain this point, at least by providing the best docking model of biotinylated M47 to CRY1, and by comparing it with M47.

We generated homology based modeled CRY1-PHR domain using RaptorX web server to fill missing amino residues. We then performed MD simulation to 300 ns. We then docked M47 each frame of the primary packet of CRY1-PHR domain. The best pose is presented in Figure 2D with binding energy of -11.2 kcal/mole. We docked the biotinylated M47 (bM47) to the CRY1-PHR and results are given in Figure S3E displayed a very similar pose with M47. The following changes were made in the text:

“The docking pose of M47 showed that it binds to the “FAD-binding” region of the CRY1 by interacting with Arg293, Trp292, Ser396, and Trp399 amino acid residues (Fig. 2E). Indoline groups of M47 interact with Trp292 through a strong pi-alkyl interaction and with Arg293 through van der Waals interaction (Fig. 2E). Additionally, benzofuran moiety of the M47 interacts with indole groups of Trp292 and Trp399 (Fig. 2E) through pi type interactions.”

5. Regarding the orientation of W292/W310 and W399/W417, the provided model (Figs. 3D and 3E) is incompatible with newer crystal structures of CRY1 and CRY2 apo forms (PDB IDs: 5T5X, 6KX4, 7D0M, and 7D0N) (for example, “CRY1” 5T5X and 6KX4 structures are similar to the “CRY2” model that the authors provided), as well as many compound-bound structures including CRY2-KL001 (PDB ID: 4MLP) and CRY1-KL001 (PDB ID: 7DLI) (i.e., the proposed orientation of the tryptophan residues interferes compound interaction in both CRY1 and CRY2). In addition, 50 ns simulation is too short to make conclusion,

compared to the standard of the field, and can be largely affected by the initial structure. Together with the point mentioned in comment 4, the provided M47 binding mode and the mechanism of CRY1 selectivity are questionable. Mutation of the key pocket residues R293 and W399 could simply disrupt the FAD pocket to prevent M47 interaction (Fig. 2E). A recent extensive study of the orientation of these tryptophan residues and its role in compound selectivity (Miller et al., PNAS 118: e2026191118, 2021) would be informative.

The effect of the mutants on CRY1 activity was assessed by transactivation assay on *Per1* promoter and the result was given in the text as follow:

“Before measuring the effect of M47 on this CRY1 mutant, we confirmed CRY1-R293A-W399L mutant retained its repressor activity with *Per1-dLuc* assay using BMAL1/CLOCK in the presences of the mutant and wild type CRY1 (Fig. S2A).”

We reanalyzed the selectivity of M47 for CRY1 using computational tools. The new data are presented in Figure 3C and D. The results were reflected in the text as follows:

“To understand why CRY2 was unable to bind M47 we performed 300 ns MD simulation of CRY1-PHR and CRY2-PHR (Fig. S3A). We superimposed the best binding conformation of M47 with CRY1 to most crowded cluster representative of CRY2. There was a steric clash between M47 and CRY2 due to different orientations of Trp292 and Ser396 (Fig. 3C). MD analyses results were also in agreement with previously published studies (Fig. S3B-E)³⁹. Additionally, conformation of these amino acids in CRY2 unfavored interaction with M47. To probe the orientation these amino acids in entire simulations of CRY1 and CRY2, χ_1 dihedral angle of Trp292 (Trp310 in CRY2) and Ser396 (Ser414 in CRY2) were calculated. The distribution of these dihedral angle was about -65° conformation for Trp310 in CRY2 while Trp292 in CRY1 was trans conformation (Fig. 3D). Ser396 of CRY1 exhibited major peak around -60° but Ser414 of CRY2 showed multiple alternate conformers at -180° , -60° , 60° , and 180° .”

6. To enable comparison with cellular assay data, please provide compound concentration in micro molar, but not ng/ml, in Figs. 5G and 5H, and Table 1. What is the expected concentration of M47 in the liver at 50 mg/kg injection in Fig. 6? Is it comparable with cellular assay data in Fig. 4 (by the way, M47 concentration information is missing in Fig. 4). Also, it is important to show CRY2 blot in Fig. 6 (and also Fig. 7A) as a control which is not affected by M47.

It is possible to give both in vivo and in vitro data as μM , but it is not possible to make a one-to-one correlation even though the units are the same. It needs to take a totally different approach to correlate in vitro and pharmacokinetics¹, which out of scope of this work. Antibody that used for the detection of the CRY2 was not working properly on samples obtained from the mice for WB. The data that we provide in Figures 1,2, and 3 are clearly show M47 is CRY1 specific.

1. Emami, J. In vitro - in vivo correlation: from theory to applications. *J Pharm Pharm Sci* 9, 169-189 (2006).

7. The specificity of M47 on CRY1 was mainly focused on the comparisons between CRY1 and CRY2. There is not enough evidence to exclude other off-targets beyond CRY2. Given Fig.2C as an example, M47 does not change the period length in CRY1 deficient cells, but the intensity (or maybe the amplitude) seems to be reduced. It would be worthwhile to consider also conducting an experiment to determine that period length and amplitude of luciferase reporter is similar to what is seen in Fig.1B in CRY2^{-/-} Bmal1-dLuc cells. Considering M47 is not affecting the cell viability and LUC stability as shown in this study, what are the other targets causing the reduced intensity in CRY1 deficient cells? In addition, it is insufficient to use Cry1^{-/-} Cry2^{-/-} MEF Per2-dLuc bioluminescence (Fig.1C) to demonstrate M47'S specificity on CRY.

To show the specificity of the M47 to CRY1 we used different strains of the U2OS cells, which are U2OS::*Bmal1-dLuc*, U2OS::*Bmal1-dLuc CRY1* knockout, U2OS::*Bmal1-dLuc CRY2* knockout, and U2OS::*Bmal1-dLuc CRY1 CRY2 double* knockout. M47 had effect on circadian rhythm in wild type and U2OS::*Bmal1-dLuc CRY2* knockout cells. However, its disappeared on CRY1 and double knockout cells.

For the concern of its effect on amplitude the following text was added to the manuscript to address the reviewer concern:

“We also observed that M47 caused not only period shortening in these cell lines, but also a dose-dependent decrease in amplitude of rhythms. These results are consistent with previous studies where either reduction or deletion of the *Cry1* genes result in short-period and low-amplitude rhythms or even arrhythmicity in different cells including U2OS and mutant mice²⁹⁻³⁴. This may be due to fact that *Cry1* is essential for generation of cell autonomous circadian clock function³⁵. To eliminate the possibility that M47 might affect the other proteins and, in turn, circadian rhythmicity we tested the effect of M47 in *CRY1/CRY2* double knockout (DKO) U2OS *Bmal1-dLuc* (Fig. S1E).

Results are given in the text as follows:

“We also used *CRY2* knockout U2OS *Bmal1-dLuc* cell (Fig. S1E) to assess selectivity of M47 on CRY1. In agreement with CRY degradation assay, M47 treatment increased the period length of U2OS *CRY2^{-/-} Bmal1-dLuc* cells in a dose dependent manner (Fig. 2D)”

Experiments such as biotinylated-M47 pull-down mass spectrometry analysis would be necessary to evaluate the off-targets in a broader scope.

Additionally, we performed pull down assay using biotinylated M47 as bait. The results are given in Table 1. M47 binds to a very few proteins in the presence of the competitors. Results are given in the text as follows:

“We used bM47 for the pull-down study using protein extract prepared from U2OS cell line in the presence and absence of competitor (M47). After pulldown, resin used for the LC-MS/MS analysis as described by Hirota et al¹⁶. The results are given in Table 1. The bM47 binds to CRY1 and its interaction decreased in the presence of competitor”

8. One selection criterion for candidate molecules in this study was the cell viability in U2OS

cells. It may not be appropriate to assess the toxicity in tumor cells such as osteosarcoma. Molecules toxic to tumors but nontoxic to normal cells are good for cancer therapy. The cell viability criteria could be done in normal healthy cells.

We included toxicity data for healthy cells using primary mouse skin fibroblast cell line (Fig S2C). Results indicate that it is not toxic at dose range that we used throughout the studies and mentioned in the text as follow:

“We also tested the toxicity of the M47 on primary mouse skin fibroblast cells, where M47 had no toxic effect (Fig. S1C)”

9. To demonstrate that the CRY1 specific effects of M47 on apoptosis and survival were not due to any other off-targets, it is better to include a CRY1^{-/-} p53^{-/-} control in studies of Fig.7.

We obtained the cell lines from another group. Unfortunately, they have no *Cry1^{-/-} p53^{-/-}* cell line. To generate such cell line requires a lot of resources and time. On the other hand, we have already showed the effect of the molecule is through CRY1 using different approach. These results clearly show M47 only change circadian rhythm in U2OS cell line. Additionally, we now include mass-spec data on off target effect of the M47 (Table 1).

10. CRY1 expression oscillates and the half-life of M47 is only some hours. In toxicity in vivo studies, the M47 injection time was 3hrs after light onset. In the survival study, M47 was injected 17hrs after light onset. Are the effects and toxicity of M47 time-dependent? Please justify the choice of different time points for M47 injection with necessary supporting evidence.

The toxicity studies with M47 in animal were being carried out independent than ZT of animals since we are interested in general toxicity of molecule. However, in survival study, it is known that the CRY1 level peaks at ZT22 therefore, we choose to administer M47 at ZT17 to reduce total CRY1 abundance in tissues considering the half-life of M47 as 6 hours. to clarify this following was added in manuscript:

“To test that 50 mg/kg/day M47 was intraperitoneally administered to male *p53^{-/-}* mice for 5 days per week at ZT17, for 7 consecutive weeks (n=9) along with vehicle-treated control animals (n=7). It known that the CRY1 level peaks at ZT22⁴⁴. To minimize the level CRY1 abundance M47 was administered at ZT17 to reduce total CRY1 abundance in tissues considering the half-life of M47 as 6 hours.”

11. Animal experiments were sometimes done in both male and female mice, sometimes in females only, and sometimes the sex was not mentioned. Please clarify the sex of animal used in each experiment and specify whether there are sex effects if both sexes were used.

The dose-escalation toxicity studies (from 5 to 1000 mg/kg m47) in mice were performed in both sex because M47 first time administered to the animals. We did not observe a sex-dependent toxicity differences in the dose escalation studies by observing clinical signs, body weight loss, body temperature and mortality according to the dose. The repeated dose toxicity studies also performed in only male mice. Subacute toxicity studies and pharmacokinetic studies were carried

out with only female mouse to not the sacrifice many animals for ethical reasons. We prefer female mice for pharmacokinetic experiments because the kinetic character of M47 would firstly assessed in the animals. In this step, we clearly demonstrated that M47 absorbed well after i.p. administration (the per oral absorption of M47 is zero percent, data not shown in the manuscript).

Additionally, only 2 mice were used per hour of treatment the brain exposure experiment. A minimum of 3 animals should be used.

Results are presented with 3 animals in Figure 5H.

12. The authors concluded in the end that “Our results suggest that reducing the level of CRY1 protects mice from tumor burden and increases the survival rate of p53^{-/-} mice”. Although the survival rate was shown, no experimental evidence regarding the protection from tumor burden was demonstrated in the results. Does M47 decrease the tumor or any histological changes after the injection of M47? Specific evidence supporting the conclusion should be presented.

We revised our conclusion as follow in the manuscript.

“Further work is needed to provide mechanistic explanations for the reduction of cancer risk in p53^{-/-} mice by M47 treatment.”

13. The cleaved PARP WB method was used for evaluating the apoptosis. But WB is not a reliable quantitative assay. It would be better to include additional more reliable quantitative assays for apoptosis.

A new data was introduced in Figure 7B. where we measured the Caspase- 3 activity. The following was added in the manuscript:

“We further show apoptosis in these cell line by measuring Caspase-3 activity. The p53-null cells treated with M47 exhibited increased Caspase-3 activity under increasing concentration of the oxaliplatin compared to cells treated with DMSO (Fig. 7B). On the other hand, wild type MSF cells had comparable Caspase -3 activity under different concentration of oxaliplatin.”

14. Authors state that the effect of M47 and selective targeting of CRY1 via M47 is promising dependent on p53 mutation status. No experiments were done to compare effects of treatment of M47 in p53^{+/+} cells or mice versus p53^{-/-} cells or mice.

As we already addressed in pervious concerns. We provide new data presented in 7B. In this study we have tested the effect of the M47 on p53^{+/+} versus p53^{-/-} cells. The description of the results is given in the text as follow:

“We further show apoptosis in these cell line by measuring Caspase-3 activity. The p53-null cells treated with M47 exhibited increased Caspase-3 activity under increasing concentration of the oxaliplatin compared to cells treated with DMSO (Fig. 7B). On the other hand, wild

type MSF cells had comparable Caspase -3 activity under different concentration of oxaliplatin.”

Additionally, it would be worthwhile to see how treatment of M47 changes tumor suppressor gene expression, BMAL1::CLOCK downstream gene expression, and cell proliferation in both cancer cells and tumor tissue in mice.

The suggested works are excellent and needs to be done as follow up studies. We are planning carry out these and other potential studies in future.

15. Fig.3A: The input portion of the blot is separate from the other conditions. This should be one single blot.

A new blot on the same gel is replaced with old one.

Other minor points:

1. Is “data not shown” allowed?

“data not shown” replaced with data Figure 1B.

2. Lines 59-60: Full names of BMAL1 and CLOCK were not written out in the introduction prior to the acronyms.

Full names are written as circadian locomotor output cycles caput (CLOCK), and aryl hydrocarbon receptor nuclear translocator-like (BMAL1) in introduction section.

3. Line 109: It would be helpful to those not familiar with the structure of CRY1 to indicated that the FAD binding pocket is located in the PHR domain, which is why the screen was designed to find molecules that target the PHR domain. The introduction only mentioned the FAD-binding pocket. Can move the sentence in Lines 119-121 up to help with this.

Lines moved to the top the section to enhance the clarity as follow as suggested by the reviewer:

“Cryptochromes (CRYs) have two distinct domains called PHR and extended C-terminal domains²⁶. PHR domain is further divided into two subdomains called primary (known to be important for FAD binding in photolyase) and secondary (important for the binding of the secondary pigment e.g. MTHF in photolyase) pockets, are critical for protein-protein interactions. For example, primary pockets, also called “FAD binding domain” is critical for the interaction FBXL3, an important ubiquitin ligase for the degradation of CRY1^{7,8,23}.”

4. Line 136: Why destabilized Luciferase was used, instead of standard Luciferase? Also, it is not clear whether the authors used dLUC or LUC in Figs. 2A and 2B. Should all LUC be dLuc (i.e. Figs S2)? If not, why were different luciferases used?

In all experiments we used destabilized luciferase. We made corrections in figure 2 legends. Following were added to the text to increase clarity with proper reference.

“Therefore, the effects of these molecules on the half-life of CRY1 were determined using CRY1::dLUC degradation assay¹⁶ where CRY1 was fused with destabilized Luciferase (dLUC) at the C-terminal previously described by Rosensweig²⁸.”

5. Some of the colors of the curves are hard to distinguish between different samples, for example, Fig.2A, B, and E are all blue and green lines, very similar. Better to use another color scheme. Also, in Fig.1C, what’s the red line? The line patterns in the plot are not consistent with the legend.

We generated the plots using different colors in Fig 2A, B and F. We further clarified related figure legends. Figure 1 is replaced with new data

6. Fig. S2D: Period change in NIH3T3 Per2-dLuc cells is not clear. Quantified period data need to be provided. In addition, NIH3T3 Bmal1-dLuc was mentioned in the main text (line 145), but the data was not provided.

Calculation of the period is provided in Fig.S1E. This figure is cited within the figure. We used NIH3T3 *Bmal1-dLuc*, we made correction in the figure legend.

7. Authors do not explain what is attributing to the lower body weight in animals given 50 mg/kg M47 at Day 2 and 3 compared to animals given vehicle or 5 mg/kg M47 (Fig.3B).

The body weight loss was observed after 50 mg/kg i.p. dose and the data very consistent with MTD study and subacute study with 60 mg/kg i.p. dose. This may have resulted from body temperature decrement of fluctuations after administration of M47 and this affects the feeding of the animals.

8. Fig.4A, maybe the blot is not a representative one, but the CLOCK protein blots intensity does look higher with M47 than with DMSO.

The new blot is replaced with old one.

9. Line 254: “next hat”?

Correction in the sentence was made as “gradually increased in the dose of 300 mg/kg (Fig. 5A)”

10. Fig. 5A shows that body temperature is higher at 5 mg/kg, while the main text says 50 mg/kg but not 5 mg/kg is higher (lines 255-257).

Following change was made in the text:

“The body temperature of the animals were higher to control groups at the dose of 5 mg/kg while the body temperatures of the animals were comparable in animals treated with 50 mg/kg M47”

11. Line 260: Should be changed to SDT

The correction was made as follows:
“All these results suggested that SDT of 5 and”

12. Line 283: Fig 4G should be Fig 5G?

Fig. 4G is replaced with Fig. 5G

13. Line 286: Remove h from “haccording”.

“h” is removed

14. Line 293: Remove th from “6th”.

“th” is removed

15. Fig. 6A legend, if not statistically significant, the null hypothesis (no difference) is accepted, and thus it is inappropriate to say “M47 slightly decreased the CRY1 levels in whole cell lysate (WCL) and cytosolic fraction”.

The statement was modified as follow in the text:
“We observed comparable CRY1 levels in the whole cell lysate of mouse liver cells (Fig 6A). Since CRYs are stabilized and degraded differently in cytosolic and nuclear compartments”

16. Figs. 6B (cytosolic) and 6C (nuclear) are incorrectly cited in the main text (lines 298-301).

The correction was made as follow in the text:
“The cytosolic levels of CRY1 in M47 treated mice liver were not statistically significant as compared to controls (Fig. 6B). On the other hand, CRY1 abundance in the nucleus was significantly low as compared to controls in dose-dependent manner (Fig. 6C).”

17. Do authors have an explanation as to why there isn't a dose dependent decrease in Per2 mRNA levels in Fig. 6D when there is a dose dependent decrease in CRY1 levels in Fig. 6C?

Although the level of the CRY1 reduced in a dose dependent manner in both concentrations, we believe the effect of the M47 was saturated at 25 mg/kg and, therefore we didn't observe dose dependent effect on Per2 transcription.

18. It is inappropriate to use t-test for the experiment with more than 2 groups (Figs. 1, 2, 4, 6).

Figure 1 we corrected the figure to indicate the t-test were done between two samples

Figure 2 we corrected the figure to indicate the t-test were done between two samples

Figure 4A we already used two-way ANOVA for statistical analyses. The information was included in the figure legend.

Figure 6 we corrected the figure to indicate the t-test were done between two samples.

Figure 6 we compared each data point with respect to its control and analyzed by t-test. We already used two-way ANOVA for statistical analyses with exception of PER2, in which each time point was compared to determine their level. This explanation was already given in figure legend.

19. Please indicate when the injection was started in Fig.7B.

Arrow is included in Fig. 7C (it was Fig. 7B previously)

20. The degree sign is not correct. Many of them are number 0, not a degree sign. When Kelvin (K) is indicated, a degree sign should not be included.

We made corrections throughout the manuscript.

21. Lines 323-326: The two sentences are contradicting each other and are suggesting that C57BL/6J mice develop cancer at 20 weeks of age regardless of p53 status. Please make sure the information of the age and cancer incidence is correct.

There were redundant two sentences and one of the is deleted.

22. Line 329: error symbols between NF and B.

Corrected as “NF- κ B”

23. Line 358: Full name of ROR was not written out prior to the acronym.

Corrected as “retinoic acid receptor-related orphan receptors”

24. Line 433: “04” needs to be removed along with the extra spaces.

“04” is removed

25. Line 468: The “s” in sec does not need to be capitalized.

corrected as “10 sec x 3 cycles”

26. Line 528: centrifugation 01 13000rpm?

corrected as “at 13000rpm”

27. Line 582: please indicate the cell number and plate size.

The cell number and plate size are given as follow:
“ 3×10^6 10 cells HEK293T cell in 10 cm plate”

28. Line 602: Indicate number of cells used.

Corrected as 5×10^5 HEK293T

29. Line 609: remove + sign.

“+” is removed.

30. Please check discrepancies between h and hr (i.e. Line 608: “hr”).

“hr” is replaced with “h”

31. Lines 286 and 701: “2nd” and “4th” should be changed to 2 h and 4 hr.

“2nd” and “4th” changed to 2 h and 4 h

32. Lack of a space after h in many “hafter” and “hof”.

correction was mad as follow:

“48 h after” and “28 h of transfection”, and “42 h of post”

33. Some “in vitro” and “in vivo” are not italicized, while others are. Please keep consistent.

“in vitro” and “in vivo” were italicized throughout manuscript.

34. Some of the letter “l” in “Luc” is not capitalized, while most are. Please fix.

All “l” replaced with “L” throughout manuscript for the Luc

35. Ref. #14 and #35 are the same. Ref. #17 and #40 are the same. Ref. #31 and #52 are the same. Ref. #47 and #56 are the same.

All references were corrected.

Reviewer #2 (Remarks to the Author):

This is an interesting paper in which the authors report the identification of a molecule from a library of approximately 2 million small molecules. This compound, M47, was among the 200 molecules in the library that exhibited computationally high affinity to the FAD binding pocket of mouse CRY1. The authors first demonstrated that M47 is not toxic to U2OS cells at physiologically relevant concentration. Then, they went on to demonstrate that M47 lengthened the period of circadian rhythm as determined by luciferase assay in derivatives U2OS and NIH3T3 cells. The drug exerts its activity by physically binding to CRY1, but not to CRY2. The authors present reasonable computational/structural evidence for this selectivity despite the fact that the “FAD binding pocket” of the two CRYs have nearly identical sequences. Finally, after determining the toxicity and pharmacokinetic profile of the drug in mice the authors explore the chemotherapeutic potential of the compound by testing it in p53^{-/-} MEF and in p53^{-/-} mice. They find that it promotes apoptotic death of the p53^{-/-} Ras transformed MSF, and increase the lifespan of p53^{-/-} mice by 25%. The authors suggest that M47 is a potential candidate as an adjuvant in cancer chemotherapy. Overall, the data that combine all state-of-the-art approaches to identify novel anticancer drugs is convincing and the paper is a valuable contribution to the field. I have the following comments that need to be addressed in the revision:

1) In the introduction the authors note that M47 destabilizes CRY1 and increase the period

length. This is contrary to Cry1^{-/-} mice. The authors, much later, in the Discussion section remark that in cell lines other small molecules that stabilize CRY1 in fact shortened period length (Ref 25) and thus, in cellulo effects do not necessarily replicate the Cry1 mutation effect in whole animal. This explanation should be given early on so the reader will not wonder to the very end about this apparent paradox.

The following statement was added to introduction section to enhance the clarity:

“A small molecule, KL001, increased the stability of CRYs and suppressed gluconeogenesis¹⁶. Additionally, stabilization of CRYs with this molecule results in lengthening of the circadian period at cellular level¹⁸. Other studies with KL001 derivatives revealed that they stabilize CRYs, but they shorten the circadian period length¹⁸.”

2) “FAD binding pocket”: This is defined by the crystal structure of photolyase [Park HW (1995) Science 268:1866]. In mammalian CRYs this pocket is wider and open to the solvent and for this and other structural reasons the mammalian CRYs have no FAD. Hence, either this point should be explicitly stated or the phrase should be in quotation marks “FAD binding pocket”.

As suggested by the reviewer FAD binding pocket used in the quotation marks throughout the manuscript.

3) Line 137: “data not shown” should be shown in supplementary Figures (Tables).

We introduced a proper reference.

4) Line 234: “slight change in BMAL1 level”: It does not look small in Fig 4A. Could it be downregulation CRY1 leads to upregulation of NR1D1 which leads to downregulation of BMAL1.

Thanks for the reviewer for its suggestion. We now incorporated following in the manuscript:

“This possibly due to the reduction of CRY1 level increases the Rev-Erb α , which is a repressor of BMAL1 expression. Increased level of the Rev-Erb α might cause the reduction of BMAL1 expression.”

5) Lines 303-304: “In agreement...decreased CRY1...[lead] to increased expression level of Per2...” is in accord with the finding of Vitaterna et al (1999) PNAS 96: 12115 which reports this effect first. This confirmatory data strengthens the authors’ agreement. I am surprised the authors are not marking this comparison.

Following revision was made in the text:

“In agreement with decreased CRY1 level in the nucleus, the expression level of *Per2* significantly increased in mice treated with M47 (Fig. 6D), which in agreement with a previously published work³³. All these suggested that M47 is effective on CRY1 levels in mouse liver cells.”

6) Lines 312-317 and Fig 7A: there is a confusion between MSF and MEF. To my knowledge

Ras24 transformed p53 null cells are MSF. This should be clarified with an appropriate reference. I could not find a mention of this in the M&M section.

We added following in M&M section:

“Cleaved PARP level was measured as described in Ozturk et al. ¹³. Briefly, Ras transformed p53-null MSF cells were grown overnight and treated with (0-2.5-5 μ M) M47 for 24 hours. Then cells were treated with oxaliplatin (0-10-20 μ M) for 16 h. Then cleaved PARP level was measure using anti-PARP (Cell Signaling). Beta-Actin levels were detected as loading control and used for the normalization of cleaved PARP signals between the samples.”

7) Minor point: At the highest concentration M47 seems to have a drastic effect on AMPLITUDE as well. A comment is needed.

Following was added to the manuscript to address the reviewer’s concern:

“We also observed that M47 caused not only period shortening in these cell lines, but also a dose-dependent decrease in amplitude of rhythms. These results are consistent with previous studies where either reduction or deletion of the *Cry1* genes result in short-period and low-amplitude rhythms or even arrhythmicity in different cells including U2OS and mutant mice ²⁹⁻³⁴. This may be due to fact that *Cry1* is essential for generation of cell autonomous circadian clock function ³⁵. To eliminate the possibility that M47 might affect the other proteins and, in turn, circadian rhythmicity we tested the effect of M47 in *CRY1/CRY2* double knockout (DKO) U2OS *Bmal1-dLuc* (Fig. S1E).”

8)The authors, in the introduction and at various sections of the paper mention clock disruption as carcinogenic and refer to some general reviews on the clock. In fact two exhaustive reviews published this year critically discuss the lack of convincing evidence for this claim. As a matter of fact this manuscript is further evidence that clock disruption is anticarcinogenic. So why make a blank statement for the claim to the contrary?

We agree with reviewer. We eliminated contradicted views from the manuscripts.

9) The writing contains numerous unusual sentences constructs commonly seen with non-native English-speaking authors. Perhaps this can be corrected by the editorial staff.

We let a native speaker read the manuscript to eliminate language problem.

Reviewer #3 (Remarks to the Author):

The Manuscript by Seref Gul et al is well written and structured with clear and logical progression throughout the MS. Here, the author describe, in detail, the discovery of a cryptochrome 1 targeting capacity and thoroughly assess its capacity to destabilize this key regulatory molecule. A point-by-point assessment of this process is seen from in vitro to In vivo approaches and in particular assess it capacity in a cancer- like setting with emphasis on P53-/- setting. While the authors assess this initially in fibroblast...which do no represent cancer as such they nicely show in a p53-/- whole body setting that this expands life and since p53 is commonly lost in cancer this claim is valid and well carried out. I feel the MS is

ready for publication with a few key aspects addressed.

Can the authors discuss the use of cell line in the study as many have p53 mutation rather than loss or intact Wt p53 and this is a critical caveat of the study...as GOF p53 is a key driver of cancer metastasis the author should at least discuss this and add these details to their study – for example USOS cells, NIH 3T3 cells, HEK 293 cells (WT for p53) thus activation of apoptosis is intact...can the authors perform experiments in a p53^{-/-} cancer cell lines in similar fashion to there initial experiments in fibroblast..... they can do this in a wt versus null setting and it is important to know if this works in GOF mutant P53 settings.

A new data was introduced in Figure 7B. where we measured the Caspase- 3 activity in both WT and null p53 in the presence and absence of M47. The following was added in the manuscript:

“We further show apoptosis in these cell line by measuring Caspase-3 activity. The p53-null cells treated with M47 exhibited increased Caspase-3 activity under increasing concentration of the oxaliplatin compared to cells treated with DMSO (Fig. 7B). On the other hand, wild type MSF cells had comparable Caspase -3 activity under different concentration of oxaliplatin. All these results suggest that M47 enhances the apoptosis in p53-null MSF cell lines.”

REVIEWER COMMENTS

Reviewer #1 (Remarks to the Author):

Comment 1:

In new Fig. 2C, CRY1/CRY2 DKO U2OS Bmal1-dLuc intensity data need quantification. In addition, statistical analysis in comparison with wild type data is required to support the authors' claim that "M47 didn't affect the luminescence dose-dependently in the absence of the CRYs".

In Fig. S1E, top panel is still labeled as "NIH3T3 Per2-dLuc". The effect of M47 on the circadian period of NIH3T3 (~0.2 h; Fig. S1E) is very small compared with U2OS cells (~2.5 h; Fig. 1B). Is there any cell type-specificity of M47 effect?

It is difficult to logically follow the sentences "We observed that M47 caused not only period lengthening in these cell lines, but also a dose-dependent decrease in amplitude of rhythms. These results are consistent with previous studies where either reduction or deletion of the Cry1 genes results in period change and low-amplitude rhythms or even arrhythmicity in different cells including U2OS and mutant mice 18,28-32.", because Cry1 knockout causes period shortening, but not lengthening. Please address this point.

Comment 2:

The authors' explanation is interesting, but is there enough space in the FAD-binding pocket to accommodate both M47 and FBXL3 C-terminal tail to form pi-pi interaction between M47 indole and FBXL3 Trp428? The authors need to provide a superposition model of CRY-FBXL3 and CRY-M47 showing there is no clash among CRY FAD-binding pocket, M47, and FBXL3 C-terminal tail to support their hypothesis.

Comment 4:

In the binding model of bM47 and M47 (new Fig. S2), there is a LARGE shift between bM47 and M47 (a shift of benzene ring size), which cannot be mentioned as "very similar pose". This shift is likely caused by avoiding a clash between the FAD-binding pocket and the biotin linker of bM47, again making the binding pose questionable as mentioned in the original comment.

Comment 5:

It is difficult to see the clash between Trp310 and M47 in new Fig. 3D. A rotated view needs to be added to help better understanding. In addition, there is a large difference in Trp292/Trp310 dynamics between new Fig. 3C and ref 38. The latter reported that CRY1 Trp292 is flexible between "up" and "down" positions, based on the CRY1 crystal structures (PDB IDs 7D0M and 5T5X/6KX4, respectively), as mentioned in the original comment.

Based on new Fig. 3C, Ser414 seems to be flexible, which can easily avoid a clash with M47. Therefore, it is not clear why Ser414 can be involved in the selectivity mechanism. These points regarding Comments 4 and 5 need to be fairly discussed.

Minor point 4:

References 16 and 64 cited in the Methods section use Luc but not dLuc for CRY degradation assays. It sounds unusual to use dLuc (which has PEST sequence for faster degradation) to evaluate the stability of CRY. Again, the authors need to confirm which luciferase was used for CRY degradation assays. Still, both dLuc and Luc are inconsistently used for CRY degradation assays in the main text and figures, and therefore confusing.

Minor point 18:

Again, t-test is not appropriate for the experiment with more than 2 groups. The authors need to use one-way ANOVA with post hoc test.

Reviewer #2 (Remarks to the Author):

The authors have addressed my concerns satisfactorily and have performed additional experiments and analyses to address the issues raised by the other reviewers. I am satisfied with the revised manuscript.

Reviewer #3 (Remarks to the Author):

Updated response answers my concerns
I am happy for publication

RESPONSE TO REVIEWER COMMENTS

Reviewer #1

Comment 1:

In new Fig. 2C, CRY1/CRY2 DKO U2OS Bmal1-dLuc intensity data need quantification. In addition, statistical analysis in comparison with wild type data is required to support the authors' claim that "M47 didn't affect the luminescence dose-dependently in the absence of the CRYs".

We have quantified the bioluminescence intensity in Fig 1C (CRY1/CRY2 DKO U2OS Bmal1-dLuc cells) and performed one-way ANOVA analysis followed by Tukey's post-hoc test (Figure 1C, Bar graph). Despite a one-way ANOVA test showing that M47 treatment decreased the bioluminescence, multiple comparison tests revealed that this effect is not dose dependent. For example, there is no significant differences among test groups treated with 1.25, 5, 10 μ M M47. In addition, we analyzed the same data by using non-linear regression model for the dose response. Normally, if an agonist or inhibitor acts on a system, dose-response analysis results in a sigmoidal curve representation. Our analysis of bioluminescence data of CRY DKO U2OS Bmal1-dLuc cells did not show such an effect (Figure 1C Right panel) and the best fitted curve had a R^2 value of 0.55. Smaller R^2 value is the indication of low fitting of the curve and absence of dose response. As a control we analyzed the period length data of WT U2OS Bmal1-dLuc cells treated with various doses of M47 given in Figure 1B (right panel). Best fitted curve had R^2 value of 0.89 which shows a strong correlation of M47 dose effect on WT U2OS Bmal1-dLuc cells. Overall, our analysis showed that there was no dose dependent effect on CRY DKO cells treated with M47.

The following revision was made in the manuscript:

"Results indicated that although all doses of M47 lowered the overall bioluminescence in the absence of the CRYs, the observed effect is not dose dependent, suggesting that the effect of M47 was through CRY"

In Fig. S1E, top panel is still labeled as "NIH3T3 Per2-dLuc".

Reviewer is right: In Fig. S1E, "NIH3T3 Per2-dLuc" in top panel is replaced with "NIH3T3 Bmal1-dLuc"

The effect of M47 on the circadian period of NIH3T3 (~0.2 h; Fig. S1E) is very small compared with U2OS cells (~2.5 h; Fig. 1B). Is there any cell type-specificity of M47 effect?

Since the M47 molecule increases the period length in both cell lines, we think its effect isn't cell specific. Since cell lines belong to human and mouse, such differences may be due to the genetic difference between them.

Following was added into manuscript:

“However, the effect of M47 was smaller on the period length of NIH 3T3 compared to U2OS. This may be due to mouse cell line having different genetic backgrounds human U2OS cell line.”

It is difficult to logically follow the sentences “We observed that M47 caused not only period lengthening in these cell lines, but also a dose-dependent decrease in amplitude of rhythms. These results are consistent with previous studies where either reduction or deletion of the Cry1 genes results in period change and low-amplitude rhythms or even arrhythmicity in different cells including U2OS and mutant mice 18,28-32.”, because Cry1 knockout causes period shortening, but not lengthening. Please address this point.

We revised the sentence as the following: “We observed that M47 caused not only period lengthening in these cell lines, but also a dose-dependent decrease in amplitude of rhythms. Decrease in amplitude is consistent with previous studies where either reduction or deletion of the Cry1 genes results in low-amplitude rhythms or even arrhythmicity in different cells including U2OS and mutant mice 18,28-32”.

Comment 2:

The authors’ explanation is interesting, but is there enough space in the FAD-binding pocket to accommodate both M47 and FBXL3 C-terminal tail to form pi-pi interaction between M47 indole and FBXL3 Trp428? The authors need to provide a superposition model of CRY-FBXL3 and CRY-M47 showing there is no clash among CRY FAD-binding pocket, M47, and FBXL3 C-terminal tail to support their hypothesis.

We believe our suggest mechanism plausible, we cannot provide data due to computational limitations. Therefore, we revised our proposed mechanism in the light of literature in the text as follow:

“Some small molecules are defined as “molecular glues” that enhance protein-protein interactions via binding the interface through different mechanisms of actions between various types of proteins³⁸. Ubiquitin ligases and their substrate interfaces are also within targets of so-called molecular glues, therefore facilitating these substrates poly-ubiquitination and proteasomal degradation³⁹⁻⁴¹. Therefore, M47 might act as a glue molecule between CRY1 and FBXL3 and in turn stabilize the interaction between them and may cause hyper ubiquitination. However, the exact mechanism in terms of the interactions between them can be obtained from the crystal structure of the CRY1-FBXL3-M47.”

Comment 4:

In the binding model of bM47 and M47 (new Fig. S2), there is a LARGE shift between bM47 and M47 (a shift of benzene ring size), which cannot be mentioned as “very similar pose”. This shift is likely caused by avoiding a clash between the FAD-binding pocket and the biotin linker of bM47, again making the binding pose questionable as mentioned in the original comment.

To assess the binding mode difference between M47 and bM47 to the primary pocket of CRY1 we measured the distance between the center of mass of bM47 and M47 atoms (excluding the biotin group). The difference between these two binding modes is 1.6 Å (single bonds between C-C atom distance is ~1.4 Å). Distance between diagonal carbon atoms of a benzene ring is ~2.7 Å, which is larger than the actual shift. As it can be seen in the **Fig. 1 (in below)**, close-up view of bM47, the biotin group still has enough space to have rotational freedom, which can be adjusted to allow deeper binding. However, docking has a caveat in this situation. Autodock Vina, and docking algorithms in general, requires a search space in the coordinate system that defines the binding area, a ligand, and a receptor. When we define bM47 as ligand Vina also searches for a binding conformation that maximizes binding affinity of biotin to the primary pocket (search space). However, there is not a proper way of excluding biotin atoms' interactions while docking and yet considering their occupancy at the same time. This is most likely why we observe this shift as an effect of including the biotin group. Overall, both docking positions are quite similar binding models with ~1.6 Å difference and help to explain our biochemical results in polyubiquitination with M47 (Fig. 2G), pull-down with bM47 (Fig 3A), and mass-spec analysis with bM47 (Table 1).

Fig. 1: Close-up view of bM47(wheat color) docked CRY1 primary pocket (blue, surface representation)

Following was added into text:

“We docked bM47 on the primary packet of CRY1 and found that it has a similar binding mode compared to M47 with a slight shift. We measured the distance between the center of masses M47 and bM47 (excluding the biotin atoms) as 1.6 Å (Fig. S2E).”

Comment 5:

It is difficult to see the clash between Trp310 and M47 in new Fig. 3D. A rotated view needs to be added to help better understanding.

We included a new data with different angle in Fig. 3D. The new data is also clearly indicated TRP310 in CRY2 clashes with M47. As can be seen in figure below shows sphere representation of M47 with Trp-310 of CRY2 clashes (**Fig. 2 in below**).

Fig.2: Two sphere superimpose images from two different angles of CRY1 and CRY2 structures. M47 binding residues (Trp399, Trp292, Arg293, and Ser396) in CRY1 and corresponding residues in CRY2 (Trp310, Ser414)

In addition, there is a large difference in Trp292/Trp310 dynamics between new Fig. 3C and ref 38. The latter reported that CRY1 Trp292 is flexible between “up” and “down” positions, based on the CRY1 crystal structures (PDB IDs 7D0M and 5T5X/6KX4, respectively), as mentioned in the original comment.

We added following sentence in the manuscript:

“Our χ_1 dihedral angle distribution Trp-292 of CRY1 is similar to previously published MD results of CRY1 (PDB ID:), where 5T5X structure was used as initial structure⁴².”

As also can be seen in **Fig 3 (in below)**, our χ_1 dihedral angle (Fig. 3A) is very similar to published data by Miller et al. (2021) (Fig. 3B). χ_1 dihedral angle distribution in Fig. 3C is quite different from the other two. This is probably due to the initial configuration of 7D0M, obtained with cryoprotectant (PG-4). This affects initial configuration of Trp-292 and χ_1 dihedral angle distribution throughout MD simulations. In fact, they also stated in the original publication (Miller et al. 2021) as follow: “Overall, the conformations of these residues in the crystal structure correlated to the predominant orientations in MD simulations, suggesting minimal effect of PG4 on the conformations of H355, H359, W399, and L400 in CRY1-PG4 with the exception of W292.”

Fig.3: χ_1 dihedral angle distributions of A) our simulations B) 5T5X from Miller et al. (2021) and C) 7D0M from miller et al. (2021) simulations.

Based on new Fig. 3C, Ser414 seems to be flexible, which can easily avoid a clash with M47. Therefore, it is not clear why Ser414 can be involved in the selectivity mechanism.

The following was added into text to increase the clarity of the conclusion:

“Although Ser-414 of CRY2 causes steric clash in the representative frame, χ_1 dihedral angle distribution shows that it has multiple conformers. Therefore, it may avoid steric clashes with M47. All these results suggest that the predominant amino acids involved in selectivity of M47 binding to CRY1 is due to the different conformations Trp-292 and Trp-310 in CRY2.”

These points regarding Comments 4 and 5 need to be fairly discussed.

Two paragraphs were revised:

“We docked bM47 on the primary pocket of CRY1 and found that it has a similar binding mode compared to M47 with a slight shift. We measured the distance between the center of masses M47 and bM47 (excluding the biotin atoms) as 1.6 Å (Fig. S2E).”

“To understand why CRY2 was unable to bind M47 we performed 300 ns MD simulation of CRY1-PHR and CRY2-PHR (Fig. S3A). We superimposed the best binding conformation of M47 with CRY1 to the most crowded cluster representative of CRY2. There was a steric clash between M47 and CRY2 due to different orientations of Trp-310 and Ser-414 (Fig. 3C). To probe the orientation of these amino acids in entire simulations of CRY1 and CRY2, χ_1 dihedral angle of Trp292 (Trp310 in CRY2) and Ser396 (Ser414 in CRY2) were calculated. The distribution of these dihedral angles was about -65° conformation for Trp310 in CRY2 while Trp292 in CRY1 was at trans conformation (Fig. 3D). Our χ_1 dihedral angle distribution Trp-292 of CRY1 is similar to previously published MD results of CRY1, where 5T5X structure was used as initial structure (Fig. S3B-E)⁴². Ser396 of CRY1 exhibited major peaks around -60° but Ser414 of CRY2 showed multiple alternate conformers at -180° , -60° , 60° , and 180° . Although Ser-414 of CRY2 causes steric clash in the representative frame, χ_1 dihedral angle distribution show that it has multiple conformers. Therefore, it may avoid steric clashes with M47. All these results suggest that the predominant amino acids involved in selectivity of M47 binding to CRY1 is due to the different conformations Trp-292 and Trp-310 in CRY2.”

Minor point 4:

References 16 and 64 cited in the Methods section use Luc but not dLuc for CRY degradation assays. It sounds unusual to use dLuc (which has PEST sequence for faster degradation) to evaluate the stability of CRY. Again, the authors need to confirm which luciferase was used for CRY degradation assays. Still, both dLuc and Luc are inconsistently used for CRY degradation assays in the main text and figures, and therefore confusing.

Reviewer is right: We changed “CRY1- dLUC and CRY2-dLUC” into “CRY1- LUC and CRY2-LUC” throughout the manuscript.

Minor point 18:

Again, t-test is not appropriate for the experiment with more than 2 groups. The authors need to use one-way ANOVA with post hoc test.

We still believe t-test is suitable, but we performed one-way ANOVA with post hoc test for Figure 1B, Figure 2A, C, and D. The changes were made in figure legend and text.

REVIEWERS' COMMENTS

Reviewer #1 (Remarks to the Author):

The authors have addressed my concerns adequately.

They should update lines 166-168 of the main text by following the explanation in the rebuttal letter: "Results indicated that although all doses of M47 lowered the overall bioluminescence in the absence of the CRYs, the observed effect is not dose dependent, suggesting that the effect of M47 was through CRY". In addition, legends to Figures 2C, 2G, 4B, 6D, S2C, and S2D may need to be updated, in which they describe t-test instead of ANOVA.

RESPONSE TO REVIEWERS' COMMENTS

Reviewer #1 (Remarks to the Author):

The authors have addressed my concerns adequately.

They should update lines 166-168 of the main text by following the explanation in the rebuttal letter: "Results indicated that although all doses of M47 lowered the overall bioluminescence in the absence of the CRYs, the observed effect is not dose dependent, suggesting that the effect of M47 was through CRY". In addition, legends to Figures 2C, 2G, 4B, 6D, S2C, and S2D may need to be updated, in which they describe t-test instead of ANOVA.

We would like to thank the reviewer. The changes were done in the manuscripts as follow:

"Results indicated that although all doses of M47 lowered the overall bioluminescence in the absence of the CRYs, the observed effect is not dose dependent, suggesting that the effect of M47 was through CRY (Fig. 1C)."

Figure Legends 2C

"(Data represent the mean \pm SEM, n=3 with duplicates **: p < 0.005, wild type U2OS *Bmal1-dLuc* versus CRY1 KO U2OS *Bmal1-dLuc* by one-way ANOVA; ns=statistically not significant one-way ANOVA in CRY1 KO U2OS *Bmal1-dLuc* DMSO vs M47 treatment)."

Figure Legends 2G

"The ubiquitination of the CRY1 in the presences of M47. The results are the average of 3 biological replicates. (One way-ANOVA *p<0.05)"

Figure Legends 4B

"(*: p < 0.05 versus DMSO control by two-way ANOVA). At 24 and 28th hours M47 increased *Per2* (Two-way ANOVA **: p<0.01, ***:p<0.001)"

Figure Legends 6D

"(Data represent the mean \pm SEM, n=3 **: p < 0.01 versus control by one-way ANOVA)"

Figure Legends S2C

"Period data is reported as the mean \pm SEM, n=2 with duplicates ***: p < 0.001 **: p < 0.005, * : p < 0.05 versus DMSO control by one way ANOVA."

Figure Legends S2D

"Data represent the mean \pm SEM, n=3 with triplicates ***: p < 0.001 versus DMSO control by one way ANOVA."